# Axonal chemokine-like Orion induces astrocyte infiltration and engulfment during mushroom body neuronal remodeling

Ana Boulanger [1,5✉], Camille Thinat[1], Stephan Züchner[2], Lee G. Fradkin[3], Hugues Lortat-Jacob [4] & Jean-Maurice Dura [1,5✉]

The remodeling of neurons is a conserved fundamental mechanism underlying nervous system maturation and function. Astrocytes can clear neuronal debris and they have an active role in neuronal remodeling. Developmental axon pruning of *Drosophila* memory center neurons occurs via a degenerative process mediated by infiltrating astrocytes. However, how astrocytes are recruited to the axons during brain development is unclear. Using an unbiased screen, we identify the gene requirement of *orion*, encoding for a chemokine-like protein, in the developing mushroom bodies. Functional analysis shows that Orion is necessary for both axonal pruning and removal of axonal debris. Orion performs its functions extracellularly and bears some features common to chemokines, a family of chemoattractant cytokines. We propose that Orion is a neuronal signal that elicits astrocyte infiltration and astrocyte-driven axonal engulfment required during neuronal remodeling in the *Drosophila* developing brain.

[1] IGH, Centre National de la Recherche Scientifique, Univ Montpellier, Montpellier, France. [2] Dr. John T. Macdonald Foundation Department of Human Genetics and John P. Hussman Institute for Human Genomics, University of Miami, Miami, FL, USA. [3] Department of Neurobiology, University of Massachusetts Medical School, Worcester, MA, USA. [4] Institut de Biologie Structurale, UMR 5075, University Grenoble Alpes, Centre National de la Recherche Scientifique, Commissariat à l'Énergie Atomique et aux Énergies Alternatives, Université Grenoble Alpes, Grenoble, France. [5] These authors jointly supervised this work: Ana Boulanger, Jean-Maurice Dura. ✉email: ana.boulanger@igh.cnrs.fr; jean-maurice.dura@igh.cnrs.fr

Neuronal remodeling is a widely used developmental mechanism, across the animal kingdom, to refine dendrite and axon targeting necessary for the maturation of neural circuits. Importantly, similar molecular and cellular events can occur during neurodevelopmental disorders or after nervous system injury[1–4]. A key role for glial cells in synaptic pruning and critical signaling pathways between glia and neurons have been identified[4]. In *Drosophila*, the mushroom body (MB), a brain memory center, is remodeled at metamorphosis and MB γ neuron pruning occurs by a degenerative mechanism[5–8]. Astrocytes surrounding the MB have an active role in the process; blocking their infiltration into the MBs prevents remodeling[9–12]. MB γ neuron remodeling relies on two processes: axon fragmentation and the subsequent clearance of axonal debris. Importantly, it has been shown that astrocytes are involved in these two processes and that these two processes can be decoupled[12]. Altering the ecdysone signaling in astrocytes, during metamorphosis, results both in a partial axon pruning defect, visualized as either some individual larval axons or as thin bundles of intact larval axons remaining in the adults, and also in a strong defect in clearance of debris, visualized by the presence of clusters of axonal debris. Astrocytes have only a minor role in axon severing as evidenced by the observation that most of the MB γ axons are correctly pruned when ecdysone signaling is altered in these cells. When astrocyte function is blocked, the γ axon-intrinsic fragmentation process remains functional and the majority of axons degenerate.

It has been widely proposed that a "find-me/eat-me" signal emanating from the degenerating γ neurons is necessary for astrocyte infiltration and engulfment of the degenerated larval axons[7,9,13]. However, the nature of this glial recruitment signal is unclear.

Here, we have identified a gene (*orion*), not previously described, by screening for viable ethyl methanesulfonate (EMS)-induced mutations and not for lethal mutations in MB clones as was done previously[14,15]. This allowed the identification of genes involved in glial cell function by directly screening for defects in MB axon pruning. We found that *orion[1]*, a viable X-chromosome mutation, is necessary for both the pruning of some γ axons and removal of the resulting debris. We show that Orion is secreted from the neurons, remains near the axon membranes where it associates with infiltrating astrocytes, and is necessary for astrocyte infiltration into the γ bundle. This implies a role for an as-yet-undefined Orion receptor on the surface of the astrocytes. Orion bears some chemokine features, for example, a CX₃C motif, three glycosaminoglycan (GAG) binding consensus sequences that are required for its function. Altogether, our results identify a neuron-secreted extracellular messenger, which is likely to be the long-searched-for signal responsible for astrocyte infiltration and engulfment of the degenerated larval axons and demonstrate its involvement for neuronal remodeling.

## Results and discussion

**The *orion* gene is necessary for MB remodeling.** Adult *orion[1]* individuals showed a clear and highly penetrant MB axon pruning phenotype as revealed by the presence of some adult unpruned vertical γ axons as well as the strong presence of debris (100% of mutant MBs; *n* = 100) (Fig. 1a, b, Table 1, and Supplementary Figs. 1 and 2). Astrocytes, visualized with *alrm-GAL4*, are the major glial subtype responsible for the clearance of the MB axon debris[12]. The presence of γ axon debris is a landmark of defective astrocyte function, as was described[11,12], and is also further shown in this study (Supplementary Fig. 1a–d). The unpruned axon phenotype was particularly apparent during metamorphosis (Fig. 1c–h). At 24 h after puparium formation (APF), although γ axon branches were nearly completely absent

in the wild-type control, they persisted in the *orion[1]* mutant brains, where we also observed a significant accumulation of debris (Fig. 1e, h). The number of unpruned axons at this stage is lower in *orion[1]* than in *Hr39[C13]* where the γ axon-intrinsic process of pruning is blocked (Supplementary Fig. 1e–g). In addition, the MB dendrite pruning was clearly affected in *orion[1]* individuals (Supplementary Fig. 1h–p).

**The *orion* gene encodes for a CX₃C motif-containing secreted proteins.** The *orion[1]* EMS mutation was localized by standard duplication and deficiency mapping as well as by whole-genome sequencing (Fig. 2a). The *orion* gene (CG2206) encodes two putatively secreted proteins: Orion-A (664 amino acid (a.a.)) and Orion-B (646 a.a.), whose messenger RNAs (mRNAs) arise from two different promoters (Fig. 2b–d). These two proteins differ in their N-terminal domains and are identical in the remainder of their sequences. The EMS mutation is a G to C nucleotide change inducing the substitution of the glycine (at position 629 for Orion-A and 611 for Orion-B) into an aspartic acid. The mutation lies in the common shared part and therefore affects both Orion-A and -B functions. Both isoforms display a signal peptide at their N termini, suggesting that they are secreted. Interestingly, a CX₃C chemokine signature is present in the Orion common region (Fig. 2b, c). Chemokines are a family of chemoattractant cytokines, characterized by a CC, CXC, or CX₃C motif, promoting the directional migration of cells within different tissues. Mammalian CX₃CL1 (also known as fractalkine) is involved in, among other contexts, neuron–glia communication[16–20]. Mammalian Fractalkines display conserved intramolecular disulfide bonds that appear to be conserved with respect to their distance from the CX₃C motif present in both Orion isoforms (Fig. 2c). Fractalkine and its receptor, CX₃CR1, have been recently shown to be required for post-trauma cortical brain neuron microglia-mediated remodeling in a mouse whisker lesioning paradigm[21]. We observed that the change of the CX₃C motif into CX₄C or AX₃C blocked the Orion function necessary for the MB pruning (Supplementary Fig. 3a–c, h–j). Similarly, the removal of the signal peptide also prevented pruning (Supplementary Fig. 3d, h–j). These two results indicate that the Orion isoforms likely act as secreted chemokine-like molecules. We also produced three CRISPR/Cas9-mediated mutations in the *orion* gene, which either delete the common part (*orion[ΔC]*), the A-specific part (*orion[ΔA]*), or the B-specific part (*orion[ΔB]*). Noticeably, *orion[ΔC]* displayed the same MB pruning phenotype as *orion[1]*, which is also the same in *orion[1]/Deficiency* females, indicating that *orion[1]* and *orion[ΔC]* are likely null alleles for this phenotype. In contrast, *orion[ΔA]* and *orion[ΔB]* have no MB phenotype by themselves indicating the likelihood of functional redundancy between the two proteins in the pruning process (Supplementary Fig. 4).

**Orion is required and expressed by MB γ axons.** Using the GAL4/UAS system[22], we found that expression of wild-type *orion* in the *orion[1]* MB γ neurons (*201Y-GAL4*) fully rescued the MB mutant phenotype (100% of wild-type MBs *n* = 387; see quantitation in Supplementary Fig. 3h), although wild-type *orion* expression in the astrocytes (*alrm-GAL4*) did not rescue (Fig. 1i–k and Supplementary Fig. 5a–c). *repo-GAL4* could not be used because of lethality when combined with *UAS-orion*. This supports the hypothesis that Orion is produced by axons and, although necessary for astrocyte infiltration, not by astrocytes. Both *UAS-orion-A* and *UAS-orion-B* rescued the *orion[1]* pruning phenotype indicating again a likely functional redundancy between the two proteins at least in the pruning process. Complementary to the rescue results, we found that the expression of an *orion*-targeting RNA interference (RNAi) in the MBs

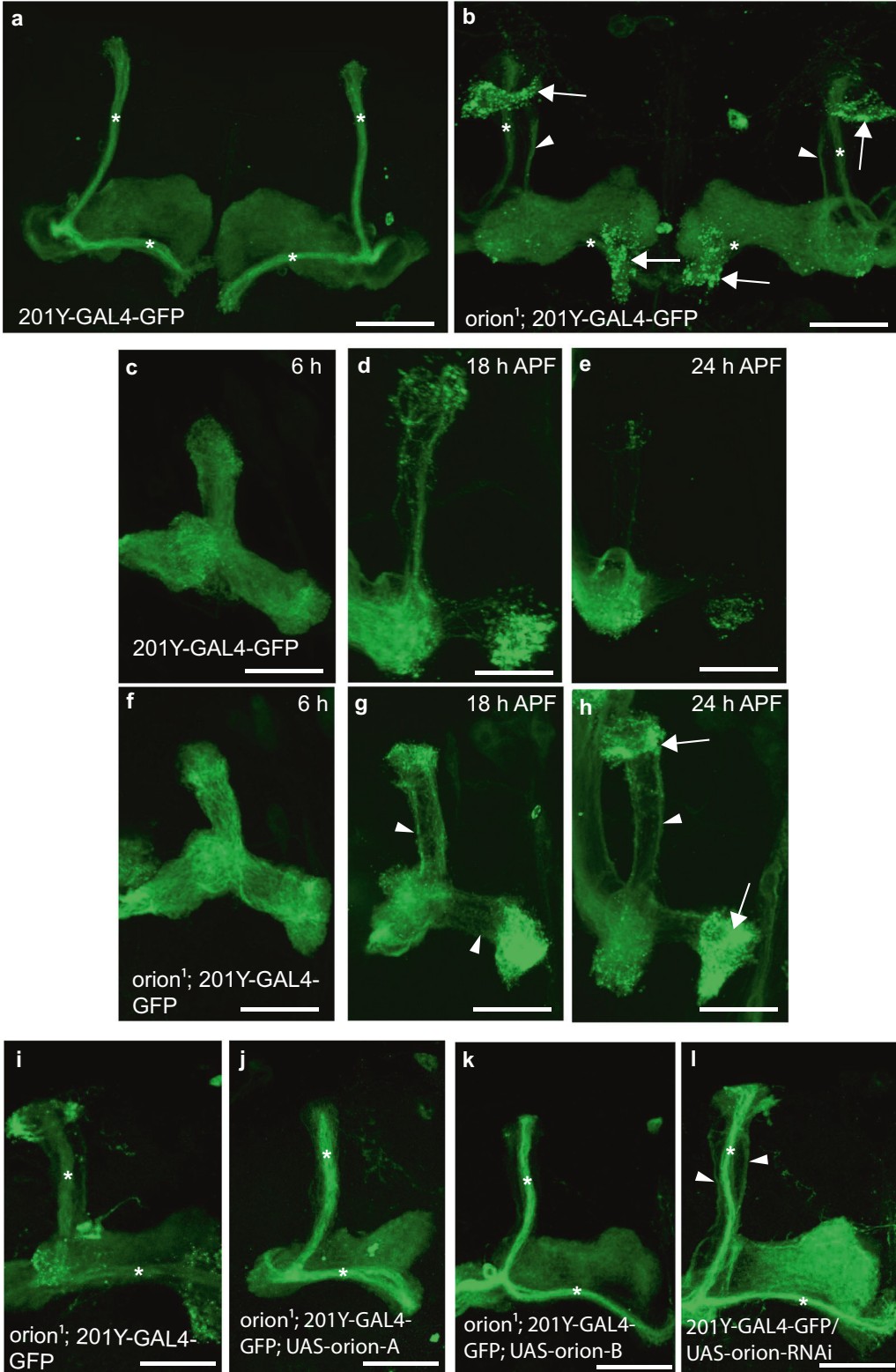

**Fig. 1 The *orion* gene is necessary for MB remodeling. a–l** γ neurons are visualized by the expression of *201Y-GAL4*-driven *UAS-mCD8-GFP* (green). In adults, this GAL4 line also labels the αβ-core axons shown here by asterisks. **a**, **b** Adult γ axons in control (**a**) and *orion[1]* (**b**). Note the presence of unpruned γ axon bundles (arrowhead) and the high amount of uncleared axonal debris (arrows) in *orion[1]* compared to wild-type. (n ≥ 100 MBs for control and *orion[1]*. See quantitation in Table 1 and Supplementary Fig. 2.) **c–h** γ axon development in wild-type (**c–e**) and *orion[1]* (**f–h**) at 6, 18, and 24 h APF as indicated. Unpruned axons (arrowhead) in *orion[1]* are already apparent at 18 h APF (compare **g** with **d**) although no differences are detected at 6 h APF (**c**, **f**). Note the presence of unpruned γ axons (arrowhead) and debris (arrow) in *orion[1]* at 24 h APF (n = 40 MBs for each developmental stage). **i–k** The adult *orion[1]* phenotype (**i**) is completely rescued by expression in MBs of *UAS-orion-A* (n = 89 MBs) (**j**) or *UAS-orion-B* (n = 387 MBs) (**k**). **l** *UAS-orion-RNAi* expression in MBs results in unpruned γ axon phenotypes (arrowheads) (n = 20 MBs). Scale bars represent 40 μm. All the images are composite confocal images. Genotypes are listed in the Supplementary list of fly strains.

**Table 1 Unpruned axon and axon debris quantitation.**

**(a) Presence of unpruned axons in ≥1-week-old adults**

| | MB | None | Weak | Strong |
|---|---|---|---|---|
| WT | 25 | 25 | 0 | 0 |
| Hr39 | 22 | 0 | 0 | 22 |
| orion$^{\Delta C}$ | 22 | 0 | 22 | 0 |
| orion$^1$ | 20 | 0 | 20 | 0 |
| orion RNAi | 34 | 0 | 34 | 0 |
| drpr$^{\Delta 5}$ | 22 | 20 | 2 | 0 |

**(b) Presence of axon debris in ≥1-week-old adults**

| | MB | None | Mild | Intermediary | Strong |
|---|---|---|---|---|---|
| WT | 25 | 25 | 0 | 0 | 0 |
| Hr39 | 22 | 22 | 0 | 0 | 0 |
| orion$^{\Delta C}$ | 22 | 0 | 0 | 0 | 22 |
| orion$^1$ | 20 | 0 | 0 | 0 | 20 |
| orion RNAi | 34 | 34 | 0 | 0 | 0 |
| drpr$^{\Delta 5}$ | 22 | 16 | 2 | 2 | 2 |

**(c) Presence of axon debris in ≤2-h-old adults**

| | MB | Scattered dots | Mild | Intermediary | Strong |
|---|---|---|---|---|---|
| WT | 10 | 10 | 0 | 0 | 0 |
| orion$^{\Delta C}$ | 12 | 0 | 0 | 0 | 12 |
| drpr$^{\Delta 5}$ | 73 | 40 | 11 | 4 | 18 |

Genotypes are indicated on the left. "MB" indicates the number of mushroom bodies observed for each genotype. Unpruned axons were ranked into three categories: "None" indicates the absence of unpruned γ axons and "Weak" and "Strong" refer to different levels of the mutant pruning phenotype. Axon debris were ranked into five categories: "None" indicates the absence of debris and "Scattered dots" means that some individual debris can be observed. "Mild," "Intermediate," and "Strong" refer to different levels of debris (see Supplementary Fig. 2 and "Methods"). Full genotypes are listed in the Supplementary list of fly strains.

functional redundancy between the two isoforms was apparent. We expressed the Orion-B protein in the γ neurons using an *UAS-orion-B-Myc* insert and the *201Y-GAL4* driver. Orion-B was present along the MB lobes and extracellularly present as visualized by anti-Myc staining (Fig. 3). Indeed, anti-Myc staining was particularly strong at the tip of the lobes indicating the presence of extracellular Orion-B (Fig. 3a, d, g, j, k). Synaptic terminals are condensed in the γ axon varicosities that disappear progressively during remodeling and hole-like structures corresponding to the vestiges of disappeared varicosities can be observed at 6 h APF[9]. We noted the presence of Myc-labeled Orion-B inside these hole-like structures (Fig. 3b, e, h). The secretion of the Orion proteins should be under the control of their signal peptide and, therefore, Orion proteins lacking their signal peptide (ΔSP) should not show this "extracellular" phenotype. When *UAS-orion-B-Myc-ΔSP* was expressed, Orion-B was not observed outside the axons or in the hole-like structures (Fig. 3c, f, i). We also excluded the possibility that this "extracellular" phenotype was due to some peculiarities of the Myc labeling by using a *UAS-drl-Myc* construct[26]. Drl is a membrane-bound receptor tyrosine kinase and Drl-Myc staining, unlike Orion-B, was not observed outside the axons or in the hole-like structures (Supplementary Fig. 6g–l). In addition, the presence of Myc-labeled Orion-B protein not associated with green fluorescent protein (GFP)-labeled axon membranes can be observed outside the γ axon bundle in 3D reconstructing images (Fig. 3j, k). Nevertheless, these signals are possibly located inside the glial compartments and not as freely diffusing Orion protein (see below). Finally, supporting the hypothesis that Orion acts as a secreted protein, it has been reported to be present in biochemically purified exosomes, indicating that it may act on the glia via its presence on or in exosomes[27].

produced unpruned axons similar to that in *orion$^1$*, although the debris is not apparent likely due to an incomplete inactivation of the gene expression by the RNAi (Fig. 1l and Supplementary Fig. 5d). The expression of the same RNAi in the glia had no effect (Supplementary Fig. 5e). Using the mosaic analysis with a repressible cell marker (MARCM[23]), we found that *orion$^1$* homozygous mutant neuroblast clones of γ neurons, in *orion$^1$/+* phenotypically wild-type individuals, were normally pruned (Supplementary Fig. 6a, b). Therefore, *orion$^1$* is a non-cell-autonomous mutation that is expected if the Orion proteins are secreted. Orion proteins secreted by the surrounding wild-type axons rescue the pruning defects in the *orion* mutant clones.

From our genetic data, *orion* expression is expected in the γ neurons. The fine temporal transcriptional landscape of MB γ neurons was recently described and a corresponding resource is freely accessible[24]. Noteworthy, *orion* is transcribed at 0 h APF and dramatically decreases at 9 h APF with a peak at 3 h APF (Supplementary Fig. 7). The nuclear receptors *EcR-B1* and its target *Sox14* are key transcriptional factors required for MB neuronal remodeling[6,7]. *orion* was found to be a likely transcriptional target of EcR-B1 and Sox14[24] and this is also consistent with earlier microarray analysis observations[25]. Noticeably, forced expression of *UAS-EcR-B1* in the MBs did not rescue the *orion* mutant phenotype and EcR-B1 expression, in the MB nuclei, is not altered in *orion$^1$* individuals (Supplementary Fig. 6c, f). Furthermore, the unpruned axon phenotype produced by *orion* RNAi is rescued by forced expression of *EcR-B1* in the MBs (Supplementary Fig. 3h). Therefore, our genetic interaction analyses support *orion* being downstream of *EcR-B1*.

**Extracellularly present Orion on MB γ axons**. We focused our further molecular and cellular work on Orion-B alone since a

**Orion is required for the infiltration of astrocytes into the MB γ bundle**. Since glial cells are likely directly involved in the *orion$^1$* pruning phenotype, we examined their behavior early during the pruning process. At 6 h APF the axon pruning process starts and is complete by 24 h APF, but the presence of glial cells in the vicinity of the wild-type γ lobes is already clearly apparent at 6 h APF[9]. We examined glial cells visualized by a membrane-targeted GFP (*UAS-mGFP*) under the control of *repo-GAL4* and co-stained the γ axons with anti-Fas2. At 6 h APF, a striking difference was noted between wild-type and *orion$^1$* brains. Unlike in the wild-type control, there is essentially no glial cell invasion of the γ bundle in the mutant (Fig. 4a–c). Interestingly, glial infiltration as well as engulfment of the degenerated larval axons was not observed in *orion$^1$* neither at 12 h APF nor at 24 h APF (Supplementary Fig. 8a–h), suggesting that glial cells never infiltrate MBs in mutant individuals. We also ruled out the possibility that this lack of glial cell activity was due to a lower number of astrocytes in mutant versus wild-type brains (Supplementary Fig. 8i, j).

We also examined the proximity between MB Orion-Myc and astrocytes, as inferred from the shape of the glial cells, labeled with the anti-Drpr antibody at 6 h APF (Fig. 4d–f). We looked at the distribution along the vertical γ lobes (60 μm of distance, see "Methods") of Orion-B-Myc (wild-type protein) and of Orion-B-ΔSP-Myc (not secreted), in an otherwise wild-type background. We quantified only from images where an astrocyte sat on the top of the vertical lobe. A peak of Orion-Myc localization was always found ($n = 10$) in the axonal region close to the astrocyte (<7 μm) when wild-type Orion-B-Myc was quantified (Fig. 4g, i). However, this was not the case ($n = 9$) when Orion-B-ΔSP-Myc was quantified (Fig. 4h, j). This strongly suggests that astrocytic processes may be "attracted" by secreted Orion.

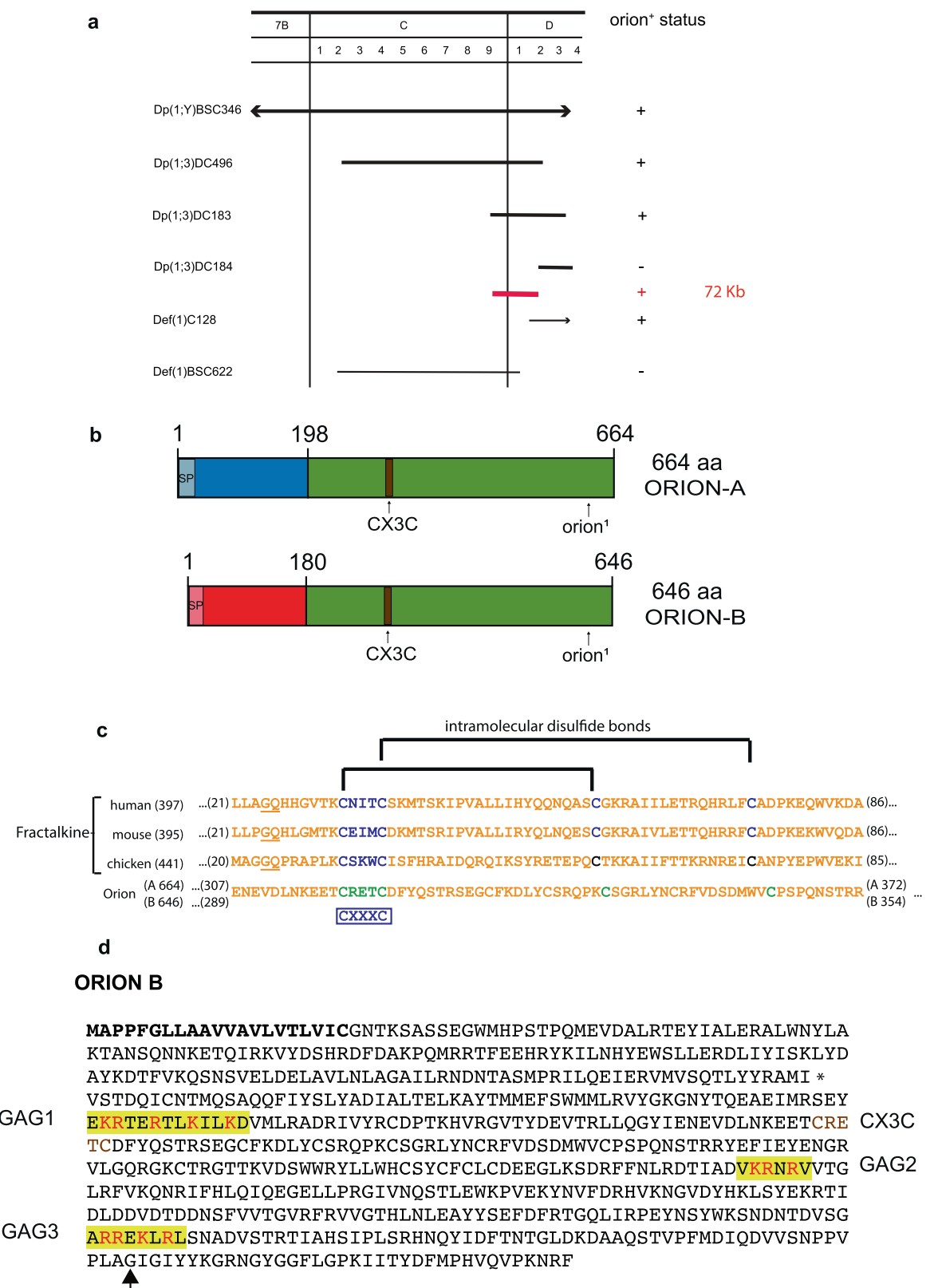

Moreover, we observed that extracellularly present Orion stays close to axon membranes (Supplementary Fig. 9a–f). Protein, in particular chemokine, localization to membranes is often mediated by GAGs, a family of highly anionic polysaccharides that occur both at the cell surface and within the extracellular matrix. GAGs, to which all chemokines bind, ensure that these signaling proteins are presented at the correct site and time in order to mediate their functions[28]. We identified three consensus sequences for GAG linkage in the common part of Orion (Fig. 2d). We mutated these sequences individually and assayed the mutant proteins for their ability to rescue the *orion[1]* pruning deficit in vivo. The three GAG sites are required for full Orion

**Fig. 2 The *orion* gene encodes for a CX₃C motif-containing protein. a** Complementation map of *orion* with the tested duplications and deficiencies in the 7B–7D region. Duplications are drawn with a heavy line and deficiencies with a light line. If *orion*[+] is present on the chromosome carrying a duplication or deficiency, it is indicated in the status column with a "+"; and if it is not present, it is marked "−." The red line indicates the location of the 72 kb to which *orion* is mapped based on the complementation results. **b** Linear representation of the polypeptide chain of the two Orion isoforms. Green represents the common region of the two Orion proteins, blue is the specific N-terminal region of Orion-A, and red the specific N-terminal region of Orion-B. The signal peptide of Orion-A and Orion-B (SP) are colored in light blue or light red, respectively. The CX₃C chemokine motif as well as the location of the *orion*[1] mutation present in the common region of Orion-A and Orion-B is indicated. **c** Amino acid sequence lineups of human, mouse, and chicken fractalkines with the common CX₃C-bearing motif of the *Drosophila* Orion proteins is shown. The number within parentheses after the species' names indicates the total length of the protein. The underlined sequences in the fractalkine sequences indicate the junctions at which their signal peptides are cleaved. The numbers at the beginning and end of the sequence indicate the protein regions in the lineup. The CX₃C (CXXXC) and conserved downstream cysteines in the fractalkine species are indicated in blue. Fractalkine intramolecular disulfide bonds between conserved cysteines[47] are specified with brackets. The CX₃C motif in the Orions and the downstream cysteines are indicated in green. The Orion downstream cysteines are offset by one and two amino acids, respectively, from those in fractalkine relative to the CX3C motif cysteines. The Orions differ from fractalkine by the inclusion of considerable extensions upstream to the CX₃C motif, while the fractalkine CX₃C motifs lie within ten amino acids of the mature signal peptide-cleaved proteins. **d** Orion-B amino acid sequence where the signal peptide is in bold, the three putative GAG binding sites (GAG1, GAG2, GAG3) are highlighted in yellow, the basic residues involved in GAG binding (R = Arg and K = Lys) are in red, and the CX₃C site is in brown. An asterisk is located at the end of the Orion-B-specific amino acid sequence/beginning of the common region. The glycine (GGC) that is mutated to an aspartic acid (GAC) in *orion*[1] is indicated by an arrow.

function, although mutating only GAG3 produced a strong mutant phenotype (Supplementary Fig. 3e–j).

Our findings imply a role for an as-yet-undefined Orion receptor on the surface of the glial cells. The glial receptor *draper* (*drpr*) seemed an obvious candidate[13,29–31], although Drpr ligands unrelated to Orion have been identified[32,33]. The MB remodeling phenotypes in *orion*[1] and *drpr*[Δ5] are, however, different with *orion* mutant phenotype being stronger than the *drpr* one. The use of an *UAS-mGFP* driven by *201Y-GAL4*, instead of anti-Fas2, where the labeling of αβ axons often masks individual unpruned γ axons, allowed us to observe occasionally unpruned axons in *drpr*[Δ5] 1-week-old post-eclosion brains in addition to uncleared debris (Table 1 and Supplementary Fig. 2), indicating a certain degree of previously undescribed unpruned axon persistence in the mutant background[13]. Nevertheless, only *orion* mutant displayed a 100% penetrant phenotype of both unpruned axons and debris (strong category) in adult flies, which are still present in old flies. On the contrary, the weaker *drpr* mutant phenotype strongly decreases throughout adulthood (Table 1 and Supplementary Fig. 1). This suggests that Drpr is not an, or at least not the sole, Orion receptor.

Independently of the possible role of Drpr as an Orion receptor, we wished to test if Orion could activate the *drpr* signaling pathway as it is the case for neuron-derived injury released factors and Spätzle5 ligands, which bind to glial insulin-like receptors and Toll-6, respectively, upregulating in turn the expression of *drpr* in phagocytic glia[31,34]. These ligands are necessary for axonal debris elimination and act as a find-me/eat-me signals in injury and apoptosis as Orion is doing for MB pruning. Our data indicate that Orion does not modify neither the Drpr expression nor the level of the *drpr* transcriptional activator STAT92E in astrocytes[35] (Supplementary Fig. 10). Consequently, Orion does not seem to induce the Drpr signaling pathway in astrocytes.

We have uncovered a neuronally secreted chemokine-like protein acting as a "find-me/eat-me" signal involved in the neuron–glia crosstalk required for axon pruning during developmental neuron remodeling. To the best of our knowledge, chemokine-like signaling in insects was not described previously and, furthermore, our results point to an unexpected conservation of chemokine CX₃C signaling in the modulation of neural circuits. Thus, it is possible that chemokine involvement in neuron/glial cell interaction is an evolutionarily ancient mechanism.

## Methods
**Drosophila stocks**. All crosses were performed using standard culture medium at 25 °C. Except where otherwise stated, alleles have been described (http://flystocks.

bio.indiana.edu). The following alleles were used: *orion*[1], *orion*[ΔA], *orion*[ΔB], and *orion*[ΔC] were generated in this study. *drpr*[Δ5rec8] was found to have an unlinked lethal mutation, which was removed by standard mitotic recombination over a wild-type chromosome[29,30]. Animals bearing this version of *drpr*[Δ5] survive to adult stages and were used for this work. The following transgenes were used: *UAS-orion-RNAi* (VDRC stock 30843) and 2× *UAS-drl-myc*[26], *10X-Stat92E-GFP*[36]. *UAS-orion-A*, *UAS-orion-A-myc*, *UAS-orion-B*, *UAS-orion-B-myc*, *UAS-orion-B-Mut AX3C-myc*, *UAS-orion-B-Mut CX4C-myc*, *UAS-orion-B-ΔSP-myc*, *UAS-orion-B-Mut GAG1-myc*, *UAS-orion-B-Mut GAG2-myc*, and *UAS-orion-B-Mut GAG3-myc* were generated in this study. We used three GAL4 lines: *201Y-GAL4* expressed in γ MB neurons, *alrm-GAL4* expressed exclusively in glial astrocytes[37] and the pan-glial driver *repo-GAL4* expressed in all glia[38].

**Mutagenesis and screening**. EMS mutagenesis was carried out following the published procedure[39]. EMS treated *y w*[67c23] *sn*[3] *FRT19A* males were crossed to *FM7c/ph*[0] *w* females and stocks, coming from single *y w*[67c23] *sn*[3] *FRT19A/ FM7c* female crossed to *FM7c* males, were generated. Only viable *y w*[67c23] *sn*[3] *FRT19A* chromosome bearing stocks were kept and *y w*[67c23] *sn*[3] *FRT19A; UAS-mCD8-GFP 201Y-GAL4/+* adult males from each stock were screened for MB neuronal remodeling defect with an epi-fluorescence microscope (Leica DM 6000).

**Mapping of orion**. To broadly map the location of the EMS-induced mutation on the X-chromosome, we used males from the stocks described in the X-chromosome duplication kit (Bloomington Stock Center) that we crossed with *orion*[1]; *UAS-mCD8-GFP 201Y-GAL4* females. Dp(1;Y)BSC346 (stock 36487) completely rescued the *orion*[1] γ axon unpruned phenotype. This duplication is located at 6D3-6E2; 7D18 on the X chromosome. We then used smaller duplications covering this region. Thus, duplications Dp(1;3)DC496 (stock 33489) and Dp(1;3)DC183 (stock 32271) also rescued the *orion*[1] mutant phenotype. However, duplication Dp(1;3) DC184 (stock 30312) did not rescue the mutant phenotype. Overlapping of duplications indicates that the EMS mutation was located between 7C9 and 7D2, which comprises 72 kb. In addition, deficiency Def(1)C128 (stock 949, Bloomington Stock Center), which expand from 7D1 to 7D5-D6, complements *orion*[1] contrarily to deficiency Def(1)BSC622 (stock 25697, Bloomington Stock Center), which does not (see Fig. 2a). We named this gene *orion* since the debris present in mutant MBs resembles a star constellation.

**Whole-genome sequencing**. Gene mutation responsible for the unpruned γ axon phenotype was precisely located through the application of next-generation sequencing. Genomic DNA was extracted from 30 adult females (mutant and control) and directly sequenced on a HiSeq2000 next-generation sequencing platform (Illumina). Bioinformatics analysis for read alignment and variant investigation was carried out through the 72 kb selected by duplication mapping (see above) at the University of Miami Miller School of Medicine, Center for Genome Technology.

**Signal peptide and transmembrane protein domain research**. For prediction of signal peptide sequences, we used the PrediSi website[40]: http://www.predisi.de; for transmembrane domains, we used the TMHMM Server, v 2.0[41]: http://www.cbs.dtu.dk/services/TMHMM/.

**Orion and fractalkine alignments**. The sequence of the region common to both Orion isoforms containing the CX₃C motifs and the likely conserved CX₃C-downstream cysteines and those of the human, mouse, and chicken fractalkine

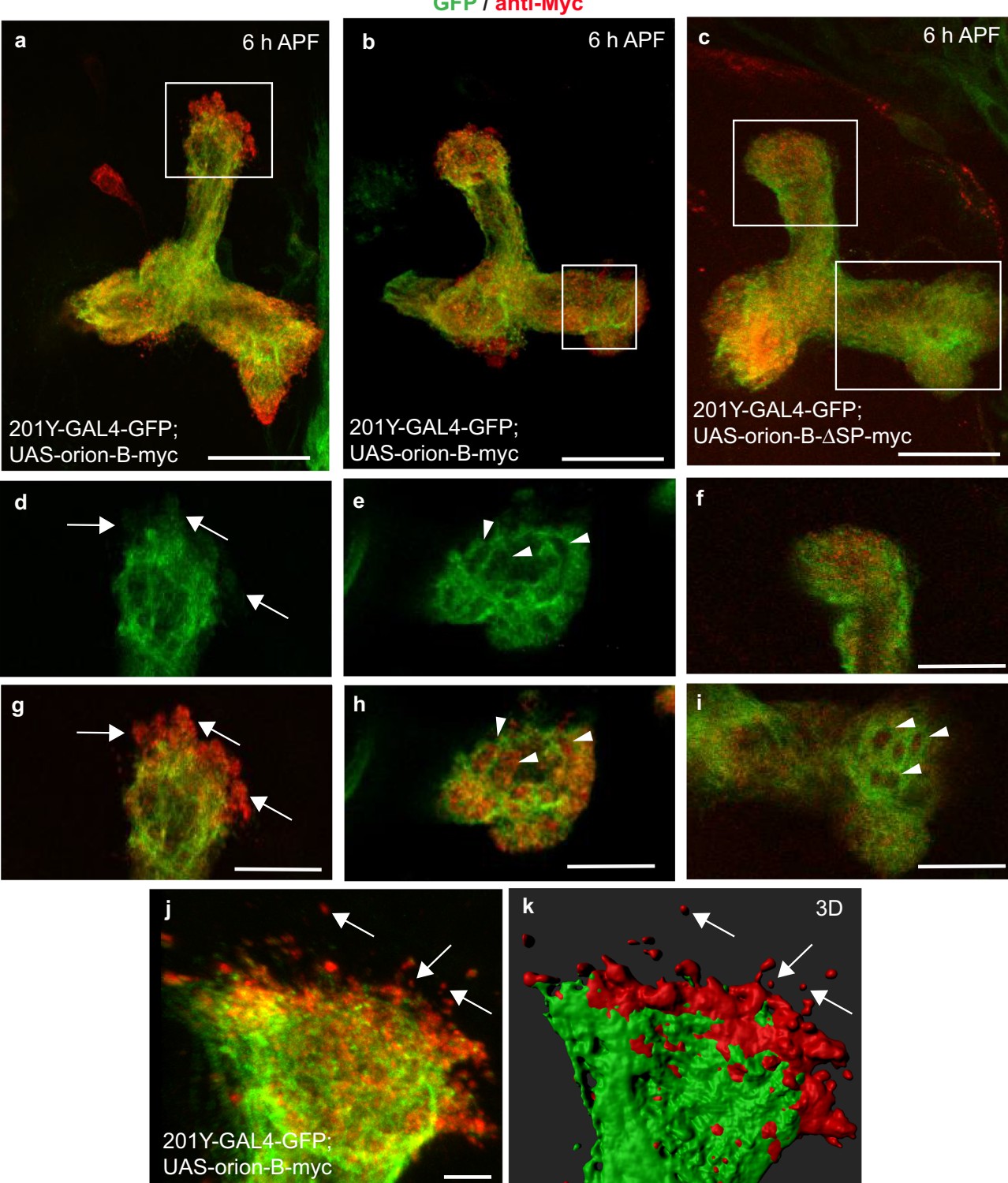

**Fig. 3 Orion is extracellularly present on MB γ axons. a–k** Six-hour APF γ axons are visualized by the expression of *201Y-GAL4*-driven *UAS-mCD8-GFP* (green). **a**, **b**, **j**, **k** γ axons expressing the wild-type Orion-B-Myc protein (red) (*n* = 10 MBs). **c** γ axons expressing the Orion-B-Myc protein lacking the signal peptide (ΔSP) (*n* = 9 MBs). **a–c** are confocal Z-projections and **j** is a unique confocal plane. **d**, **g** Higher magnification images of the region indicated by a rectangle shows a representative unique confocal plane. Note the presence of Myc-labeled Orion-B outside the γ axon bundle (arrows). **e**, **h** Higher magnification images of the region indicated by a rectangle in **b** showing a representative unique confocal plane. Note the presence of Myc-labeled Orion-B inside the hole-like structures present in the γ axon bundle (arrowheads). **f**, **i** Higher magnification images of the vertical and medial γ lobes, respectively (rectangles in **c**). Orion-B-ΔSP-Myc is observed neither outside the γ axons (**f**) nor in the hole-like structures (arrowheads in **i**). **j**, **k** Presence of Myc-labeled Orion-B extracellular proteins not associated with GFP-labeled axon membranes can be observed outside the γ axon bundle (arrows). **k** Three-dimensional surface-rendering (3D) of the confocal image. **j** Reveals close apposition of GFP-labeled axons and Myc-labeled Orion and reveals Orion is present as small extracellular globules. Scale bars represent 40 μm in **a–c**, 20 μm in **d–i** and 5 μm in **j**, **k**. Full genotypes are listed in the Supplementary list of fly strains.

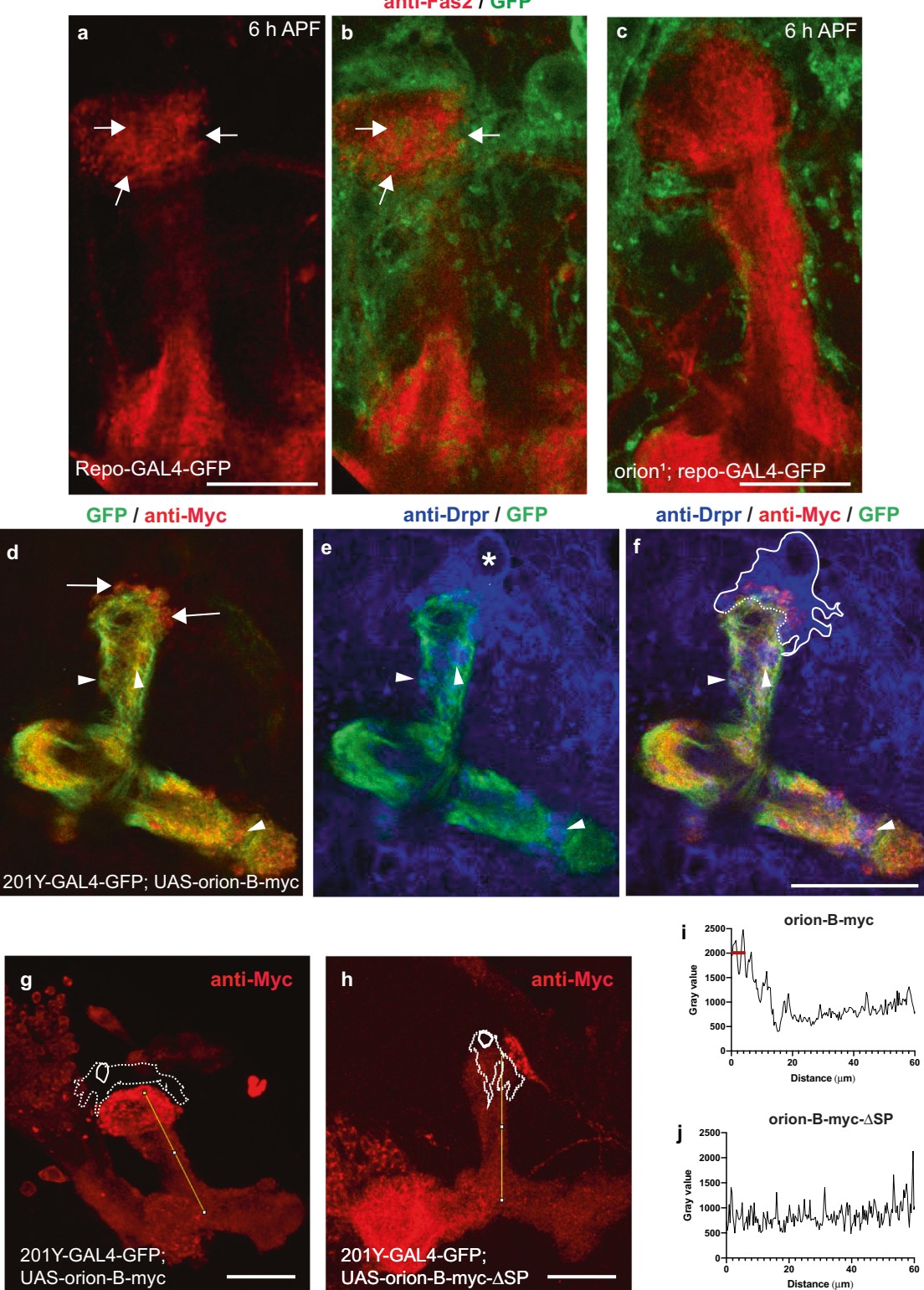

were aligned using the AlignX plug-in in the VectorNTi software package (Invitrogen) without permitting the introduction of spaces or deletions.

**GAG binding site research**. Identification of GAG binding sites in proteins, in the absence of structural data, is complicated by the diversity of both GAG structure and GAG binding proteins. Previous work based on heparin-binding protein sequence comparisons led to the proposition of two GAG binding consensus sequences, the XBBXBX and XBBBXXBX motifs (where B and X stand for basic and neutral/hydrophobic amino acids, respectively). A number of closely related basic clusters, including XBBXBXBX, were next experimentally identified[42]. Visual

examination of the Orion-B sequences returned three such clusters (XBBXXB XXBXXBX: residues 242–254; XBBBXBX: residues 416–421; and XBBXBXBX: residues 547–554, numbering includes the peptide signal, see Fig. 2d), which are also present in Orion-A.

**CRISPR-Cas9 strategy**. All guide RNA sequences (single guide RNA (sgRNA)) were selected using the algorithm targetfinder.flycrispr.neuro.brown.edu/ containing 20 nucleotides each (PAM excluded) and are predicted to have zero off-targets. We selected three pairs of sgRNA. Each pair is targeting either the A-specific region of *orion*, the B-specific region of *orion*, or the C common region of the two

**Fig. 4 Orion is required for the infiltration of astrocytes into the MB γ bundle and engulfment of the larval axons at 6 h APF. a–c** Single confocal planes of 6 h APF brains expressing *repo-GAL4*-driven *UAS-mCD8-GFP* (green) in controls (**a**, **b**) and *orion[1]* (**c**) focused on the MB dorsal lobe (*n* = 12 MBs controls and *n* = 20 MBs *orion[1]*). Anti-Fas2 staining (red) reveals spherical hole-like structures occupied by glial processes infiltrating into the γ bundle (green, arrows) in wild-type (**a**, **b**), but not in *orion[1]* individuals (**c**). Scale bars are 20 μm. **d–f** A single confocal plane showing the expression of *201Y-GAL4*-driven *UAS-mCD8-GFP* (green, **d–f**) and Orion-B-Myc (red, **d**, **f**) in 6 h APF MB. Anti-Drpr antibody (blue) was used to visualize the glial cells (blue, **e**, **f**). **d–f** display the same MB. **d** displays Orion-B-Myc expression outside the axons at the top of the vertical γ bundle (arrows) as well as in hole-like structures (arrowheads). **e** displays an astrocyte positioned at the top of the γ bundle (asterisk in its nuclei) as well as several astrocyte processes occupying hole-like structures (arrowheads). Note the colocalization of Orion-B-Myc and glia processes in the hole-like structures (arrowheads in **f**). The astrocyte cell membrane (continuous line) and the membrane contacting the tip of the γ bundle (dotted line), where phagocytosis is taking place, based on our interpretation of the astrocyte limits according to the green and the blue channels for GFP and Drpr, respectively, are indicated in **f**. *n* = 10 MBs. Scale bar is 40 μm. **g**, **h** Representative images to illustrate how the quantitation of Orion-B-Myc expression (red) (**g**) and Orion-B-ΔSP-Myc (**h**) driven by *201Y-GAL4* in a traced 60 μm line contained in a γ axon vertical bundle was performed. The position of an astrocyte (dotted line), labeled by anti-Drpr staining and its nucleus (solid circle) are indicated. **i**, **j** Representative plotted intensity profiles of Orion expression (gray value) in Orion-B-Myc- (**i**) or Orion-B-ΔSP-Myc-expressing MBs (**j**), according to the distance from the tips (0 μm) to the bottoms (60 μm) of vertical γ bundles. The highest peaks are always located at <7 μm distance to the tip of the vertical bundle (red bar) when Orion-B-Myc is expressed (*n* = 10), although this was never the case (*n* = 9) when secretion-deficient Orion-B-ΔSP-Myc expression was quantified. Scale bar in **g**, **h** are 30 μm. Source data are provided as a Source Data file. Full genotypes are listed in the Supplementary list of fly strains.

isoforms. We used the following oligonucleotide pairs: CRISPR-1 orion A fwd and CRISPR-1 orion A rev to target the A region, CRISPR-1 orion-B fwd and CRISPR-1 orion-B rev to target the B region, and CRISPR-1 orion common region C fwd and CRISPR-1 orion common region C rev to target the C region (see the corresponding oligonucleotide sequences in Supplementary Table I). We introduced two sgRNA sequences into pCFD4[43], a gift from Simon Bullock (Addgene, plasmid # 49411) by Gibson Assembly (New England Biolabs) following the detailed protocol at crisprflydesign.org. For PCR amplification, we used the protocol described on that website. Construct injection was performed by Bestgene (Chino Hills, CA) and all the transgenes were inserted into the same attP site (VK00027 at 89E11). Transgenic males expressing the different *orion* sgRNAs were crossed to *y nos-Cas9 w** females bearing an isogenized X chromosome. One hundred crosses were set up for each sgRNA pair, with up to five males containing both the sgRNAs and *nos-Cas9*, and five *FM7c/ph[0]* w females. From each cross, a single *y nos-Cas9 w* /FM7c* female was crossed with *FM7c* males to make a stock that was validated for the presence of an indel by genomic PCR with primers flanking the anticipated deletion and the precise endpoints of the deletion were determined by sequencing (Genewiz, France) using *orion*-specific primers.

**Adult brain dissection, immunostaining, and MARCM mosaic analysis**. Adult fly heads and thoraxes were fixed for 1 h in 3.7% formaldehyde in phosphate-buffered saline (PBS) and brains were dissected in PBS. For larval and pupal brains, brains were first dissected in PBS and then fixed for 15 min in 3.7% formaldehyde in PBS. They were then treated for immunostaining as described[23,44]. Antibodies, obtained from the Developmental Studies Hybridoma Bank, were used at the following dilutions: mouse monoclonal anti-Fas2 (1D4) 1:10; mouse monoclonal anti-Draper (8A1), 1:400; and mouse monoclonal anti-Repo (8D1.2) 1:10. Mouse monoclonal primary antibody against EcR-B1 (AD4.4) was used at 1:5000[45]. Polyclonal mouse (Abcam, (9E10) ab32) and rabbit (Cell Signaling, 71D10) anti-Myc antibodies were used at 1:1000 and 1:500, respectively. Goat secondary antibodies conjugated to Cy3, Alexa 488, and Cy5 against mouse or rabbit IgG (Jackson Immunoresearch Laboratory) were used at 1:300 for detection. To generate clones in the MB, we used the MARCM technique[23]. First instar larvae were heat-shocked at 37 °C for 1 h. Adult brains were fixed for 15 min in 3.7% formaldehyde in PBS before dissection and GFP visualization.

**Microscopy and image processing**. Images were acquired at room temperature using a Zeiss LSM 780 laser scanning confocal microscope (MRI Platform, Institute of Human Genetics, Montpellier, France) equipped with a ×40 PLAN apochromatic 1.3 oil-immersion differential interference contrast objective lens. The immersion oil used was Immersol 518F. The acquisition software used was Zen 2011 (black edition). Contrast and relative intensities of the green (GFP), of the red (Cy3), and of the blue (Cy5 and Alexa 647) channels were processed with the ImageJ and Fiji software. Settings were optimized for detection without saturating the signal. For each set of figure settings were constants. However, since the expression of the Orion-B-Myc-ΔSP protein is lower than the one of the Orion-B-Myc (as shown in the western blot and its quantitation Supplementary Fig. 3i–j), the levels of red were increased in this particular case in order to get similar levels than in Orion-B-Myc. We used the Imaris (Bitplane) software to generate a pseudo-3D structure of Orion-produced γ axons (Imaris surface tool). We created two 3D surfaces, from regular confocal images, defining the axonal domain (green) and the Orion expression domain (red).

**Quantitation of immunolabelling**. To quantify unpruned γ axons, we established three categories of phenotypes: "none," when no unpruned axons are observed,

"weak," when few unpruned individual axons or thin axon bundles are observed in the dorsal lobe, and "strong," when >50% of the axons are unpruned. In this last category, the percentage of unpruned axons is estimated by the width of the corresponding medial bundle compared with the width of the medial pruned axon bundle[44]. For debris quantification, we established five categories: none, scattered dots, mild, intermediary, and strong based on the location and size of the debris clusters[11]. In ≥1-week-old adults, "none" means the absence of debris. In ≤2-h-old adults "scattered dots" means some individual debris can be observed. We considered "mild," if debris clusters (clusters >5 μm$^2$) appear only at one location, "intermediary," at two locations, and "strong," at three locations of the MB. Three debris locations were considered: the tip of the vertical lobe, the tip of the medial lobe, and around the heel (bifurcation site of γ axons into dorsal and medial).

For EcR-B1 signal quantitation, we performed five measurements for each picture (intensity 1, ..., 5) in the nuclei of GFP-positive cell bodies and the same number of measurements in the background using confocal single slices. The mean of these background measurements is called mean background. We then subtracted intensities of mean background from each intensity value (intensity 1, ..., 5 minus mean background) to obtain normalized intensity values. Finally, we compared normalized intensity values between two genetic conditions. We proceeded similarly for Draper and STAT-GFP signal quantitation, but staining was quantified in the astrocyte cytoplasm located in the immediate vicinity of the γ dorsal lobe. Quantitation of intensity was performed using the ImageJ software.

To quantify the Myc signal in the γ vertical lobe, we traced a 60 μm line on the Cy3 red Z-stack and used the Plot Profile function of ImageJ to create a plot of intensity values across the line. The top of the line (0 μm) was located at the tip of the γ vertical lobe and the bottom of the line (60 μm) at the branching point of the two γ lobes. Only images containing an astrocyte sitting at the top of the γ vertical lobe were used to quantify Myc expression levels in *orion-B-* and *orion-B-ΔSP*-expressing MBs.

To quantify the number of astrocytes around the γ lobes, we counted the number of glial nuclei, as labeled with anti-Repo antibody, contained in GFP-positive astrocyte cytoplasm labeled with *UAS-mCD8-GFP* driven by *alrm-GAL4*. We only counted nuclei contained within a circle of 70 μm of diameter centered in the middle of the vertical γ lobe tips.

**UAS constructs**. The *orion-A cDNA* (complementary DNA) inserted in the pOT2 plasmid (clone LD24308) was obtained from Berkeley Drosophila Genome Project (BDGP). Initial Orion-B cDNA as well as the *Orion-B cDNAs* containing mutations at the CX$_3$C and the GAG sites or lacking the signal peptide were produced at GenScript (Piscataway, NJ) in the pcDNA3.1-C-(k)DYK vector. The *Orion-B cDNAs* contained the following mutations:

To remove the signal peptide, we removed the sequence: GCGCCGCCTTTCGGATTATTAGCTGCTGTTGTTGCTGTTCTTGTCACGCT TGTGATTTGTGGAAATA located after the first ATG. At the CX$_3$C site: in AX$_3$C we exchanged TGC to GCC. In CX4C, we added an additional Ala (GCC) to get CAX$_3$C. To mutate the putative GAG binding sites, we exchanged Lys and Arg by Ala at the corresponding sites: AAGAGGACGGAACGCACACTAAAAATACTC AAG; AAGCGCAACCGA and CGCAGGGAGAAACTGCGT to GCCGCCACGG AAGCCAACCACTAGCCATACTCAAG, GCCGCCAACGGC, and GCCGCCGAGG CCCTGGCC, respectively, for mutations in GAG1, GAG2, and GAG3. The different constructs were amplified by PCR using forward primers containing sequences CACCaaaacATG (where ATG encodes the first methionine), followed by the specific *orion-A* or *orion-B* cDNA sequences and common reverse primers. To amplify orion-A, we used primers: orion-A fwd and orion-AB rev, and to amplify orion-B as well as orion-B containing mutations, we used primers: orion-B fwd and orion-AB rev. In addition, to amplify orion-ΔSP, we used orion-B ΔSP fwd and

orion-AB rev (see the corresponding oligonucleotide sequences in Supplementary Table I),

Amplified cDNA was processed for pENTR/D-TOPO cloning (Thermo Fisher Scientific, K240020) and constructs were subsequently sequenced (Genewiz, France). We used the Gateway LR clonase enzyme mix (Thermo Fisher Scientific, 11791019) to recombine the inserts into the destination UAS vector pJFRC81-GW-2× Myc (Lee G. Fradkin, unpublished), which was generated from pJFRC81-10XUAS-IVS-Syn21-GFP-p10 (Addgene, plasmid 36462 deposited by G. Rubin[46]) by replacing the GFP ORF with a Gateway cassette adding on a C-terminal 2× Myc tag. orion-A, orion-B, and Myc-tagged constructs (orion A, orion-B, and orion-B mutants) transgenic fly lines (UASs) were generated at BestGene and all the transgenes were inserted into the same attP site (VK00027 in 89E11). All the crosses involving the UAS-GAL4 system were performed at 25 °C, except for UAS-orion-A and 201Y-GAL4, which were performed at 18 °C.

**Western analysis**. Five larval heads were homogenized in an Eppendorf tube containing 20 µl of 3× sample buffer (2% sodium dodecyl sulfate (SDS), 0.125 M Tris-HCl pH 6.9, 5% β-mercaptoethanol, 20% glycerol, bromophenol blue) and proteins were separated by SDS–polyacrylamide gel electrophoresis. After electrotransfer to nitrocellulose, the blot was blocked in PBS, 0.5% Tween-20, and 5% milk. The Orion-Myc and tubulin proteins were detected using a mouse anti-Myc antibody (clone 9E10, Abcam) and an anti-tubulin antibody (Sigma, T5168) at 1/1000 and 1/10,000, respectively, in PBS, 0.5% Tween-20, 5% milk, and revealed using anti-mouse Ig horseradish peroxidase (1:10,000) and an ECL kit (Amersham). Band intensities were normalized to the corresponding tubulin band intensity using the ImageJ software.

**Statistics**. Comparison between two groups expressing a qualitative variable was analyzed for statistical significance using the two-sided Fisher's exact test. Comparison of two groups expressing a quantitative variable was analyzed using the two-sided nonparametric Mann–Whitney $U$ test. (BiostaTGV: http://biostatgv.sentiweb.fr/?module=tests). Values of $p < 0.05$ were considered to be significant. Graphs were performed using the GraphPad Prism software (version 8.1.1).

**Reporting summary**. Further information on research design is available in the Nature Research Reporting Summary linked to this article.

## Data availability
The authors declare that the data supporting the findings of this study are available within the paper and Supplementary information files. Source data are provided with this paper.

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

## Acknowledgements

We thank Amélie Babled, Pascal Carme, and Dana Bis-Brewer for help in the EMS mutagenesis, MB developmental studies, and WGS analysis, respectively, and Oren Schuldiner for discussions about the Orion expression and function. We also thank Marc Freeman for *alrm-GAL4* stock, Baeg Gyeong Hun for *10X-STAT92E-GFP* stock, the Bloomington *Drosophila* Stock Center and VDRC for fly stocks, the BioCampus RAM-*Drosophila* facility (Montpellier, France), the imaging facility MRI, which is part of the UMS BioCampus Montpellier and a member of the National Infrastructure France-BioImaging, supported by the French National Research Agency (ANR-10-INBS-04) for help in confocal and image analysis and processing. We acknowledge BDGP, BestGene, GenScript, and Genewiz for cDNA clone, transgene service, gene synthesis, and DNA-sequencing, respectively. The 1D4 anti-Fasciclin II hybridoma and the 8D12 anti-Repo monoclonal antibody developed by Corey Goodman and the 8A1 anti-Draper monoclonal antibody developed by Mary Logan were obtained from the Developmental Studies Hybridoma Bank, created by the NICHD of the NIH and maintained at The University of Iowa, Department of Biology, Iowa City, IA 52242. C.T. was supported by grants from the INSB at the CNRS and from the Fondation pour la Recherche Médicale. Work in the laboratory of J.-M.D. was supported by the Centre National de la Recherche Scientifique, the Association pour la Recherche sur le Cancer (grants SFI20121205950 and PJA 20151203422) and the Fondation pour la Recherche Médicale (Program "EQUIPES FRM2016" project DEQ20160334870).

## Author contributions

A.B. and J.-M.D. designed the project; A.B., C.T., and J.-M.D. performed the experiments; A.B., S.Z., L.G.F., H.L.-J., and J.-M.D. analyzed the data; A.B. and J.-M.D. wrote the original draft of the manuscript; A.B., L.G.F., H.L.-J., and J.-M.D. reviewed and edited the manuscript.

## Competing interests

The authors declare no competing interests.
