## [Peer Review File · Nature Communications]

Reviewers' Comments:

Reviewer #1:

Remarks to the Author:

The Boulanger et al. manuscript entitled, "Axon-secreted chemokine-like Orion is a signal for astrocyte infiltration during neuronal remodeling" defines a function for a very interesting neuronal cue required for pruning and engulfment of mushroom body lobe axons in *Drosophila*. The authors find that loss of Orion results in persistent intact lobe axons as well as axon debris. They show that Orion is required in the neurons themselves and that it is secreted from axon terminals. Lastly, they show that loss of Orion results in the absence of astrocytes entering the lobe. On the basis of these results, they conclude that Orion is sensed by an as-yet-unidentified astrocyte receptor. This is a logical and well-presented study that will be of significant interest to neurobiologists interested in neuron pruning and neuron-glia interactions. There are some issues that should be addressed prior to publication.

1. The overall Orion LOF phenotype appears consist of two distinct phenotypes: impaired axon breakdown (as evidenced by persistent intact axons) and a phagocytosis defect (evidenced by punctate axonal debris). Does the presence of unpruned axons in the mutants suggest that Orion may be involved in original axon fragmentation, not just glial engulfment? To this reviewer, this phenotype hints that Orion acts upstream of the neuron-glia signal entirely, and instead might be involved in a signaling interaction between lobe axons to drive initial degeneration. This possibility seems especially reasonable to consider given that it is unknown whether the Orion receptor is neuronal or glial.
2. On a related note, as presented, the authors only state that 100% of lobes are aberrant. It is important to indicate what fraction display a pruning defect in addition to the engulfment defect to enable a comparison to *drpr* LOF animals, which are not reported to display a pruning defect. In general, there was surprisingly little data quantification in the paper. Even for example, for Orion RNAi phenotypes, which were said to be weaker.
3. On p. 4, there is a reference to a missing Figure 11 when discussing an Orion RNAi phenotype.
4. The authors characterize Orion-B protein expression by overexpressing a Myc tagged transgene in lobe neurons. They convincingly show that it is strikingly localized to axon terminals. (a) To have an indication of whether this protein reflects the localization of the endogenous protein, the authors should demonstrate that it rescues the Orion LOF phenotype. (b) While the protein appears outside the axonal membrane, it does not look extracellular, but rather localized and globular. Do the authors believe that it is within astrocytic membranes? This question appears to be somewhat addressed in Figure 4F, but *Drpr* is not exclusively membrane-localized, so it is difficult to tell. Can the authors label astrocytic membranes with mCD8GFP to see if Orion-myc is engulfed? 3D renderings would also help make this point convincing.
5. The title of ms. emphasizes the function of Orion in astrocyte infiltration into the degenerating lobe. The data in support of this claim is shown in Figure 4 A-C. But again related to (1), the lobe axons in Figure 4C do not appear to be fragmented at all. It is unclear to this reviewer why the astrocytes would be infiltrating if the axon was not degenerating. It would be more convincing if the authors could show fragmenting axons that are not actively being engulfed by astrocytes in order to point for a defect in axon-glia communication.

Reviewer #2:

Remarks to the Author:

The manuscript by Boulanger et al. describes a chemokine-like molecule that mediates glial invasion during the phase of axon pruning and ingestion in the mushroom bodies of *Drosophila*.

The *Drosophila* mushroom bodies (MB) have been studied for the past 20 years as the major system for examining the developmental remodeling of neurons. The pruning phase of this remodeling is under steroid control and involves interaction between the MB neurons and the glial cells that surround them. Early in the pruning process, glial extensions invade the MB lobes and participate in both the severing of axons and the removal of axonal debris. This paper is an important advancement because it identifies a secreted protein, that they name orion, that appears to be the agent that attracts the glia into the MB lobes. In orion null mutants, the glial incursions that normally occur by 6 hr APF are not present. The authors show that myc-labeled orion protein is secreted from the MB neurons and accumulates outside of the neural cell membranes and that this secretion does not occur if a putative signal peptide is deleted from the orion gene.

Overall, the data supplied by the authors nicely support their claims. There are two things that would improve their paper. The first is that it is clear that in the absence of orion, the glia do not remove the debris that results from axon degeneration (e.g., Fig 1) and that the magnitude of the effect seems equivalent to the lack of Draper (Fig. S1C), a glial receptor involved in phagocytosis of neuronal remains. Importantly, though, Orion and Draper do not seem to be parts of the same pathway. What I found troublesome is that authors' claim that Orion is necessary for the pruning of the axons of the gamma cells. Indeed, there are some persistent axons in orion mutants but the axon survival is not near what they show for hr39 mutants (Fig s1D). The benchmark for the suppression of pruning would be similar to that seen after removal of steroid signaling as seen in EcR null mutants. (e.g., Lee et al., 2000, Neuron). Their 24 hr APF image (Fig 1H) does not show axon survival that is anything close to that seen at the corresponding time point for EcR MARCM clones from the Lee paper. At best, it appears to have a minor role in axon severing. Indeed, if orion were necessary for pruning there would be no debris to remove in these mutants.

The second point is that the paper would be improved by a better documentation of the interaction between the MB neurons and glia through the first 24 hr APF – the time during which the neurons are being pruned and fragments removed. We are shown images of MB neurons at 6, 18 and 24 hr APF [Fig 1] but their relationship to the surrounding glia is only shown at 6 hr [Fig 4]. A better time course including labeled glia would show whether the lack of orion prevents glial incursion or only delays it. It might also shed some light into why a few axons survive.

Minor issues:

To avoid confusing the non-*Drosophila* MB specialists, the authors should state that 201Y-GAL4 expression in the adult is seen in both gamma and alpha-beta Kenyon cells, a feature that should be indicated on Fig 1A.

Figure S1K. It is difficult to discriminate the persistence of dendrites (*) from the degeneration profiles.

Figure 4. In 'G' the authors show the image used to generate the profile presented in the first part of part H [for orion-B-myc]. They should also show the image that was the basis of the preparation that lacks the signal sequence.

Reviewer #3:

Remarks to the Author:

Drosophila mushroom body (MB) neuron is an excellent model system to study developmental neural remodeling. Larval axons of *Drosophila* MB neurons degenerate at the early pupal stage. The degenerated axons are phagocytosed by astrocytes. However, it is unknown how degenerating axons recruit astrocytes. In this study, by unbiased genetic screen, Bolanger et al identified a new molecule, orion, which is essential to recruit astrocyte. In the homozygous orion mutant, astrocytes do not engulf degenerated axon, resulting in remaining axonal debris in adult brain. Interestingly, orion encodes molecule containing the CX3C (CxxxC) chemokine signature and are secreted from degenerating axons. Basing on these findings, the authors claim that Orion is the

neuronal signal that elicits astrocyte infiltration required for developmental neuronal remodeling. This claim is very interesting because such a chemokine has not been yet identified in *Drosophila*. However, the authors have not showed enough experimental data to support their claim. Based on the existence of the CX3C motif and a N-terminal signal peptide, they assume this molecule would work as a chemokine-like ligand. Although they claimed that Myc-tagged Orion expressed in MB neurons was secreted to outside of MB lobe, their interpretation on this result is apparently wrong (see below). The authors should perform more definitive experiments to support their claim. Because this journal is one of high profiling journals, readers expect high impact study with clear and definitive experiments. My opinion is that the conclusion of this study will fascinate readers, but this study lacks enough data to prove their claims. Please refer to following comments. I have several concerns on this study.

Major points:

1. The authors expect orion encodes chemokine-like molecule because of existence of CX3C chemokine signature. Do all molecules containing CX3C have chemokine-like function? Because this motif is composed of just 5 amino acids with two consensus, this reviewer thought it is difficult to predict the function of this molecule with just such a short motif alone. At least, please show the references showing that this short motif together with N terminal signal peptide alone is sufficient to predict function as a chemokine.

To test the requirement of the CX3C motif for function of Orion, the authors showed that the change of this motif into CX4C or AX3C could not rescue the Orion function in MB axon pruning. With this result, the authors assumed that Orion likely acts as a secreted chemokine-like molecule. However, it is difficult to understand this logic for me. Although this result showed that the essential function of CX3C motif, this doesn't mean that this molecule acts as a secreted chemokine. Please explain this connection more carefully.

2. The authors induced expression of Myc-tagged Orion in MB and found Myc signals outside of MB lobe. Based on this result, they conclude that Orion is secreted molecule. However, their judgment and interpretation on this result is apparently wrong. In the Figure 3D/G and 3E/H, the anti-Myc signals in the outside of MB (tip of MB lobe) with arrows (3D, G) and inside the hole-like structure with arrowheads (3E, F) are shown. However, apparently these anti-Myc signals are overwrapped with weak GFP signals. These weak GFP signals are localized on MB varicosities phagocytosed by astrocyte. Therefore, it is hard to believe that these anti-Myc signals were secreted from MB and located outside of MB. It has been shown that the anti-FasII signals were also localized in the outside of MB (tip of MB lobe) (Fig 4F in ref. 9) and hole-like structure (Fig 3D and 4I, J in ref. 13) in the process of pruning. These anti-FasII signals were localized on varicosities or axons phagocytosed by astrocytes. Therefore, similar to anti-FasII signals, anti-Myc signals should be located on MB axons or boutons engulfed by astrocyte.

If the Orion is actually secreted MB lobe, overexpression of the Myc-tagged Orion would be observed outside of MB lobe before engulfment of astrocytes. At 0h APF, when original orion is not expressed in MB neuron, these exogenous expressed Myc-Orion might be found outside of MB lobe if this molecule is actually secreted from MB lobe. Also, it is feasible to do other experiment to observe secretion of Myc-Orion; Because expression of EcR-DN in MB neurons suppresses axon pruning and successive phagocytosis by astrocytes, it is possible to distinguish the secreted Myc signals from phagocytosed Myc signals, if EcR-DN and Myc-Orion are expressed together in MB neurons.

3. Using the clonal analysis of orion mutant MB neurons with MARCM, the authors showed that loss of orion in MB neurons are normally cleared by phagocyte. Their interpretation on this result is that the orion $-/+$ (heterozygote mutant) MB axons secret Orion normally and recruit phagocyte to the bundle of MB axons, resulting in phagocytosis of all MB axons by infiltrated phagocyte processes. This result indicates that Orion functions to recruit glial membrane rather than as "eat-me" signals for phagocyte. This is interesting and essential point of this research. To confirm the functional aspect of Orion, they need to perform other experiments. If pruning defect in orion mutant is induce by failure of recruitment of glial membrane, ectopic expression of orion in MB

alpha'/beta' neurons would rescue the pruning defect. MB alpha'/beta' neurons extend their axons into the core of larval MB lobes and their axons are localized close to axons of MB gamma neurons.

4. It is very important to know generality of function of Orion for recruitment of glial processes. In adult fly, severed axons are phagocytosed by glial cells. Marc Freeman's group established nice model system to observe the glial function for clearance of severed axons and published a lot of papers. Interestingly, Drpr is essential for clearance of both developmentally degenerating axons and severed axons. Is Orion also required for both pupal and adult glia to phagocyte axons? I think the authors should include this experiment in this report to further discuss function of Orion.

Minor points:

1. The authors revealed that "our genetic interaction analyses support orion being genetically downstream of EcR-B1." It is hard to understand the result of "genetic interaction analysis." They showed that the overexpression of EcR-B1 in orion homozygous mutant MB neurons did not affect the MB pruning phenotype of the orion mutant. I can't understand that why this result show that orion is downstream of EcR-B1. I can't accept this logic. To show this notion, the authors should perform other experiments. If the EcR-B1 signaling induces expression orion, it is expected that the overexpression of EcR-B1 in MB neurons induces more orion expression. In this case, effect of expression of orion RNAi in MB neurons would be suppressed in some extent by the increased expression of orion.

2. Although the authors measure the signal intensity of Orion-B-Myc along the MB vertical lobe and compared with that of Orion-B-deltaSP-Myc. This reviewer can't understand the point of this experiment. I can understand the requirement of signal peptide. The authors claimed the attraction of astrocytic process by Orion with this experiment. However, it is hard to understand the logic.

Reviewer #1 (Remarks to the Author):

The Boulanger et al. manuscript entitled, "Axon-secreted chemokine-like Orion is a signal for astrocyte infiltration during neuronal remodeling" defines a function for a very interesting neuronal cue required for pruning and engulfment of mushroom body lobe axons in *Drosophila*. The authors find that loss of orion results in persistent intact lobe axons as well as axon debris. They show that orion is required in the neurons themselves and that it is secreted from axon terminals. Lastly, they show that loss of Orion results in the absence of astrocytes entering the lobe. On the basis of these results, they conclude that orion is sensed by an as-yet-unidentified astrocyte receptor. This is a logical and well-presented study that will be of significant interest to neurobiologists interested in neuron pruning and neuron-glia interactions. There are some issues that should be addressed prior to publication.

1. The overall orion LOF phenotype appears consist of two distinct phenotypes: impaired axon breakdown (as evidenced by persistent intact axons) and a phagocytosis defect (evidenced by punctate axonal debris). Does the presence of unpruned axons in the mutants suggest that orion may be involved in original axon fragmentation, not just glial engulfment? To this reviewer, this phenotype hints that Orion acts upstream of the neuron-glia signal entirely, and instead might be involved in a signaling interaction between lobe axons to drive initial degeneration. This possibility seems especially reasonable to consider given that it is unknown whether the orion receptor is neuronal or glial.

We have better explained the previously described relationship between glia and axon fragmentation in the results (lines 48-57). MB γ neuron remodeling consists of two processes: axon fragmentation and the subsequent clearance of axonal debris. Importantly, it has been shown that astrocytes are involved in these two processes and that these two processes can be decoupled (Hakim et al. 2014 - ref 12). Altering ecdysone signaling in astrocytes, during metamorphosis, results both in a partial axon pruning defect, and in a strong defect in debris clearance. Nevertheless, astrocytes have only a minor role in axon severing as evidenced by the observation that most of the MB γ axons are correctly pruned when ecdysone signaling is altered in these cells. When astrocyte function is blocked, the γ axon-intrinsic fragmentation process remains functional and the majority of axons degenerate.

The MB *orion* phenotype is similar, with partial axon fragmentation and strong axon debris clearance defects, to the one described when *EcR-B1* is blocked in astrocytes (ref 12). Particularly, the partial axon pruning defect is visualized as either some individual larval axons or as thin bundles of intact larval axons remaining in the adults and axonal debris are visualized by the presence of clusters close to the MBs. Currently, the only way to see debris is to alter the glial function either by blocking glial cell movement with *UAS-shi^{ts}* (Awasaki and Ito, 2004 - ref 9) or by blocking the required signaling pathway within the glial cells with *UAS-EcR-DN* (ref 12). Thus, the presence of debris indicates that Orion is involved in the neuron-glia crosstalk. When the MB cell-autonomous function of *EcR-B1* is blocked, the unpruned axons phenotype is highly penetrant but there is no of debris because axon fragmentation is blocked (see Supplementary Fig. 1d). *orion* function has only a minor role in

axon severing as evidenced by the observation that most of the MB γ axons are correctly pruned in *orion* mutant adults (Fig. 1b versus 1a and Supplementary Fig. 1b versus 1a and 1d). Finally, related to that issue it is interesting to read the comments of Reviewer #2 who is concerned by the same problem. That said, it is a very good hypothesis to propose that some (but clearly not all) MB axons should be fragmented under the action, not only of MB dependent EcR-B1, but also of secreted Orion in association with a neuronal receptor in an autocrine way.

2. On a related note, as presented, the authors only state that 100% of lobes are aberrant. It is important to indicate what fraction display a pruning defect in addition to the engulfment defect to enable a comparison to *drpr* LOF animals, which are not reported to display a pruning defect. In general, there was surprisingly little data quantification in the paper. Even for example, for *orion* RNAi phenotypes, which were said to be weaker.

We have added Table I and Sup Fig. 2 where both the unpruned axons and the axon debris are quantified in different genotypes including *drpr* ^{Δ 5} (lines 211-214) and *orion* RNAi. Please, see also lines 361-372 in “Quantitation of immunolabelling” in the “Methods” section. We have also included the quantitation of *UAS-orion-RNAi* and its rescue by co-expression with *UAS-EcR-B1* in Sup Fig. 3h (previous Sup Fig. 2h).

3. On p. 4, there is a reference to a missing Figure 11 when discussing an *orion* RNAi phenotype.

This is Fig. 11 (1 and “L”) not 11 (now on p.5 line 133).

4. The authors characterize Orion-B protein expression by overexpressing a Myc tagged transgene in lobe neurons. They convincingly show that it is strikingly localized to axon terminals.

(a) To have an indication of whether this protein reflects the localization of the endogenous protein, the authors should demonstrate that it rescues the *orion* LOF phenotype.

It is indeed the case. This is described page 4 lines 122-124 and in Sup Fig. 3h (previous Sup Fig. 2h).

(b) While the protein appears outside the axonal membrane, it does not look extracellular, but rather localized and globular. Do the authors believe that it is within astrocytic membranes? This question appears to be somewhat addressed in Figure 4F, but *Drpr* is not exclusively membrane-localized, so it is difficult to tell. Can the authors label astrocytic membranes with mCD8GFP to see if Orion-myc is engulfed? 3D renderings would also help make this point convincing.

In order to address this reviewer’s experiment proposal, we would need to drive *UAS-orion-B-myc* with *201Y-GAL4* and therefore would have to use a GAL4-independent system to label astrocytes with mGFP. To the best of our knowledge, this tool has not yet been used. We

were, however, able to sometimes see Orion aggregates engulfed by glia in 3D renderings of confocal images (Rebuttal Fig. 1). This suggests that Orion can indeed be localized within astrocytes.

Rebuttal Fig. 1. Secreted Orion is engulfed by glia. 3D renderings of γ axons expressing 201Y-GAL4-driven UAS-mCD8-GFP (green) and the wild type Orion-B-Myc protein (red) using the Imaris section tool, at 6 h APF. Glial cells are labeled with an anti-drpr antibody (blue). The nucleus of an astrocyte is indicated with an asterisk. Secreted Orion puncta (white arrows) are surrounded by astrocytic processes in the three spatial axes, indicate that Orion is engulfed by astrocytes. XY, XZ and YZ are confocal planes in each axis at the cross-hair point indicated (n = 5). Scale bar, 10 μ m. Genotype: $y w^{67c23} / Y$ or $y w^{67c23} / y w^*$; UAS-mCD8GFP 201Y-GAL4 / +; UAS-orion-B-myc / +.

5. The title of ms. emphasizes the function of Orion in astrocyte infiltration into the degenerating lobe. The data in support of this claim is shown in Figure 4 A-C. But again related to (1), the lobe axons in Figure 4C do not appear to be fragmented at all. It is unclear to this reviewer why the astrocytes would be infiltrating if the axon was not degenerating. It would be more convincing if the authors could show fragmenting axons that are not actively being engulfed by astrocytes in order to point for a defect in axon-glia communication.

We first want to stress that the presence of uncleared axonal debris reveals two aspects of the remodeling process in *orion* mutants. The debris observed in *orion* mutant indicate that axon fragmentation does take place. When axon fragmentation is completely blocked by removing MB *EcR-B1* function there is no apparent axonal debris. Second, the presence of uncleared debris indicates that glial cells are not functioning properly. The most obvious phenotype visualized in the *orion* mutant brain, at 6 h APF (Fig. 4c), is the lack of glial infiltration into the axon bundle. We have now extended the documentation of the interaction between MB neurons and glia to include the first 24 h APF (Supplementary Fig. 8a-h). Interestingly, glial infiltration was not observed in *orion*¹ either at 12 h APF or at 24 h APF indicating that glial cells do not infiltrate MBs in mutant individuals (lines 182-184). Nevertheless, the overall structure of the γ bundle appears similar in wild-type and mutant at 12 h APF as well as at 24 h APF. This indicates that, in the *orion* mutant, axon fragmentation takes place, as in wild type, and fragmenting γ axons are not actively being engulfed by astrocytes.

Reviewer #2 (Remarks to the Author):

The manuscript by Boulanger et al. describes a chemokine-like molecule that mediates glial invasion during the phase of axon pruning and ingestion in the mushroom bodies of *Drosophila*. The *Drosophila* mushroom bodies (MB) have been studied for the past 20 years as the major system for examining the developmental remodeling of neurons. The pruning phase of this remodeling is under steroid control and involves interaction between the MB neurons and the glial cells that surround them. Early in the pruning process, glial extensions invade the MB lobes and participate in both the severing of axons and the removal of axonal debris. This paper is an important advancement because it identifies a secreted protein, that they name orion, that appears to be the agent that attracts the glia into the MB lobes. In *orion* null mutants, the glial incursions that normally occur by 6 hr APF are not present. The authors show that myc-labeled orion protein is secreted from the MB neurons and accumulates outside of the neural cell membranes and that this secretion does not occur if a putative signal peptide is deleted from the orion gene.

Overall, the data supplied by the authors nicely support their claims. There are two things that would improve their paper. The first is that it is clear that in the absence of orion, the glia do not remove the debris that results from axon degeneration (e.g., Fig 1) and that the magnitude of the effect seems equivalent to the lack of Draper (Fig. S1C), a glial receptor involved in phagocytosis of neuronal remains. Importantly, though, Orion and Draper do not seem to be

parts of the same pathway. What I found troublesome is that authors' claim that Orion is necessary for the pruning of the axons of the gamma cells. Indeed, there are some persistent axons in orion mutants but the axon survival is not near what they show for hr39 mutants (Fig s1D). The benchmark for the suppression of pruning would be similar to that seen after removal of steroid signaling as seen in EcR null mutants. (e.g., Lee et al., 2000, Neuron). Their 24 hr APF image (Fig 1H) does not show axon survival that is anything close to that seen at the corresponding time point for EcR MARCM clones from the Lee paper. At best, it appears to have a minor role in axon severing. Indeed, if orion were necessary for pruning there would be no debris to remove in these mutants.

We completely agree with this Reviewer on this point and this is why we propose to Reviewer#1 to read this Reviewer#2's comment (Please, read our answer to Reviewer#1's comment#1). We now present the quantitation of the presence of unpruned axons and axon debris in wild-type, in *Hr39^{C13}*, where *EcR-B1* is blocked in MBs, and in *orion* mutant brains in Table I and Supplementary Fig. 2.

The second point is that the paper would be improved by a better documentation of the interaction between the MB neurons and glia through the first 24 hr APF – the time during which the neurons are being pruned and fragments removed. We are shown images of MB neurons at 6, 18 and 24 hr APF [Fig 1] but their relationship to the surrounding glia is only shown at 6 hr [Fig 4]. A better time course including labeled glia would show whether the lack of orion prevents glial incursion or only delays it. It might also shed some light into why a few axons survive.

We have now documented the interaction between neurons and glia through the first 24 h APF (Supplementary Fig. 8). Interestingly, glial infiltration was not observed in *orion*¹ either at 12 h APF or at 24 h APF indicating that glial cells apparently do not infiltrate mutant individuals MBs axon bundles (lines 182-184).

Minor issues:

To avoid confusing the non-Drosophila MB specialists, the authors should state that 201Y-GAL4 expression in the adult is seen in both gamma and alpha-beta Kenyon cells, a feature that should be indicated on Fig 1A.

This is done. We have marked the adult $\alpha\beta$ -core axons labelled by 201Y-GAL4 in all the figures where they appear with asterisks and explain this in all the relevant figure legends.

Figure S1K. It is difficult to discriminate the persistence of dendrites (*) from the degeneration profiles.

This is now addressed. Supplementary Fig. 1n was changed accordingly; a single confocal plane of a dendritic region containing larval dendrite debris (brilliant dots) has been added.

Figure 4. In 'G' the authors show the image used to generate the profile presented in the first part of part H [for orion-B-myc]. They should also show the image that was the basis of the preparation that lacks the signal sequence.

This is now done (Fig. 4h).

Reviewer #3 (Remarks to the Author):

Drosophila mushroom body (MB) neuron is an excellent model system to study developmental neural remodeling. Larval axons of Drosophila MB neurons degenerate at the early pupal stage. The degenerated axons are phagocytosed by astrocytes. However, it is unknown how degenerating axons recruit astrocytes. In this study, by unbiased genetic screen, Bolanger et al identified a new molecule, orion, which is essential to recruit astrocyte. In the homozygous orion mutant, astrocytes do not engulf degenerated axon, resulting in remaining axonal debris in adult brain. Interestingly, orion encodes molecule containing the CX3C (CxxxC) chemokine signature and are secreted from degenerating axons. Basing on these findings, the authors claim that Orion is the neuronal signal that elicits astrocyte infiltration required for developmental neuronal remodeling.

This claim is very interesting because such a chemokine has not been yet identified in Drosophila. However, the authors have not showed enough experimental data to support their claim. Based on the existence of the CX3C motif and a N-terminal signal peptide, they assume this molecule would work as a chemokine-like ligand. Although they claimed that Myc-tagged Orion expressed in MB neurons was secreted to outside of MB lobe, their interpretation on this result is apparently wrong (see below). The authors should perform more definitive experiments to support their claim. Because this journal is one of high profiling journals, readers expect high impact study with clear and definitive experiments. My opinion is that the conclusion of this study will fascinate readers, but this study lacks enough data to prove their claims. Please refer to following comments. I have several concerns on this study.

Major points:

1. The authors expect orion encodes chemokine-like molecule because of existence of CX3C chemokine signature. Do all molecules containing CX3C have chemokine-like function? Because this motif is composed of just 5 amino acids with two consensus, this reviewer thought it is difficult to predict the function of this molecule with just such a short motif alone. At least, please show the references showing that this short motif together with N terminal signal peptide alone is sufficient to predict function as a chemokine.

The reviewer raises the question as to whether the presence of a CX₃C motif on a secreted or transmembrane protein (whose precursor bears a signal peptide sequence) is sufficient to confer “chemokine-like” activity on a protein. We fully agree that the presence of a CX₃C motif does not necessarily defines a chemokine: it has been found, for example, in the mitochondrial import machinery “translocase of inner membrane” (Koehler CM Trends Biochem Sci 2004). The human respiratory syncytial virus (RSV) attachment glycoproteins, which can be expressed as membrane-anchored (mG) or secreted (sG) forms, also both contain a central fractalkine-like CX₃C motif. Interestingly, although the RSV G protein is not a chemokine, CX₃C chemokine mimicry for this protein has been reported and its binding to CX₃CR1, the specific fractalkine receptor, facilitates RSV infection of cells (Tripp RA et al. Nat Immunol 2001). Regarding Orion, we would like first to recall that fractalkine itself is an unusual chemokine: it has a high molecular mass, it is the only member of the CX₃C family and it occurs in either a soluble or a membrane associated form. *Sensu stricto*, secretion is not necessary to define a chemokine. In the present work, we described that Orion, in addition to a CX₃C motif, bears functional GAG binding sites, which are also a typical and highly conserved chemokine signature. It is exported in the extracellular milieu where it displays chemotactic/haptotactic activity. The CX₃C chemokine fractalkine has been recently shown to be involved in developmental pruning in the mammalian brain (Gunner et al., 2019 - ref 21), thus share with Orion similar biological activities. As chemokines have not been yet identify in *Drosophila* or even in insects, we agree that we have no comparison helping to firmly characterize Orion as a typical chemokine. Nevertheless, the reasons why we propose that *orion* encodes a chemokine-like molecule follow:

- 1) Orion is involved in neuronal remodeling by acting on a neuron-glia crosstalk.
- 2) Orion is secreted by neurons and is necessary for glial cell function.
- 3) Orion bears a functional CX₃C motif (as evidenced by the failure of the mutant to rescue).
- 4) Orion bears functional GAG binding motifs (as evidenced by the failure of mutant to rescue).
- 5) The Orion protein isoforms bear two cysteines carboxyterminal to the CX₃C motif at nearly identical positions to those in fractalkine where they have been shown to form intrachain disulfide bonds with the cysteines in the fractalkine CX₃C motif (now described in Fig. 2c and in lines 105-107 in the text).
- 6) The mammalian CX₃C chemokine fractalkine has recently been shown to play a similar role in developmental pruning in the brain.

Thus, given the above-described characteristics, we believed Orion can be appropriately referred as a chemokine-like protein.

To test the requirement of the CX₃C motif for function of Orion, the authors showed that the change of this motif into CX₄C or AX₃C could not rescue the Orion function in MB axon pruning. With this result, the authors assumed that Orion likely acts as a secreted chemokine-like molecule. However, it is difficult to understand this logic for me. Although this result showed that the essential function of CX₃C motif, this doesn't mean that this molecule acts as a secreted chemokine. Please explain this connection more carefully.

Orion bears a signal peptide that is necessary for its function (Sup Fig. 3d versus 3a) and whose mutation clearly prevents Orion secretion (Fig. 4g, i versus 4h, i). This underlies our hypothesis that Orion acts as a secreted protein. Orion also bears a CX₃C motif and, although Orion is not homologous to the mammalian fractalkine (the only CX₃C chemokine described), we have tested if this motif could be relevant for its function. It is indeed the case. This why we propose that secreted Orion has a chemokine-like mode of function. We have improved the description of these results in the text (lines 109-113).

2. The authors induced expression of Myc-tagged Orion in MB and found Myc signals outside of MB lobe. Based on this result, they conclude that Orion is secreted molecule. However, their judgment and interpretation on this result is apparently wrong. In the Figure 3D/G and 3E/H, the anti-Myc signals in the outside of MB (tip of MB lobe) with arrows (3D, G) and inside the hole-like structure with arrowheads (3E, F) are shown. However, apparently these anti-Myc signals are overwrapped with weak GFP signals. These weak GFP signals are localized on MB varicosities phagocytosed by astrocyte. Therefore, it is hard to believe that these anti-Myc signals were secreted from MB and located outside of MB. It has been shown that the anti-FasII signals were also localized in the outside of MB (tip of MB lobe) (Fig 4F in ref. 9) and hole-like structure (Fig 3D and 4I, J in ref. 13) in the process of pruning. These anti-FasII signals were localized on varicosities or axons phagocytosed by astrocytes. Therefore, similar to anti-FasII signals, anti-Myc signals should be located on MB axons or boutons engulfed by astrocyte.

We believe the Reviewer misunderstood the experiment we describe in Fig. 3 where we drive *UAS-orion-myc* and *UAS-mGFP* with *201Y-GAL4*. The GFP labels the axons and Myc is appended to Orion which is expressed and secreted by the same axons. We did not label the glial cells in Fig. 3. The Reviewer makes reference to experiments described in Fig 4F in ref. 9 and in Fig 3D and 4I, J in ref. 13 where glia is labelled with GFP and axons with Fas2. It is not clear to us how these two sets of data can be compared. According to the Reviewer, anti-Fas2 signals and anti-Myc signals should be similar. As we can see in our Fig. 3a, d and g and Fig. 3b, e and h, the anti-Myc signal is clearly different from the GFP signal. Nevertheless, it is largely documented that the γ neuron GFP signal and the anti-Fas2 signal are very similar at 6 h APF. See, for instance, Fig. 3B and Fig. S1E in ref. 9. These labeling are clearly different from what is shown in Fig. 3. Moreover, we observe a clear difference between wild-type Orion-Myc (Fig. 3a and b) and Δ SP Orion-Myc expression (Fig. 3c) which indicates that the Myc expression we see in Fig. 3a and b correlates with Orion secretion. Similarly, we observe a clear difference in expression between the membrane-spanning Drl-Myc receptor and Orion-Myc (Sup Fig. 6g-i versus Sup Fig. 6j-l). The association of weak membrane GFP staining with Orion-Myc is likely due to GAG binding as described in the manuscript (lines 196 – 205). Nevertheless, it is clearly possible to observe Myc-labelled Orion-B secreted protein that is not associated with GFP-labelled axon membranes outside of the γ axon bundle in 3D reconstruction images (lines 170-172, 356-538 and Fig. 3j, k). See also our answer to comment 4b of Reviewer#1.

In order to definitively settle this point and convince this Reviewer, we now present evidence that the expression pattern of Fas2 and neuronal GFP are similar (Rebuttal Fig. 2a-g) although the expression pattern of secreted Orion and Fas2 are clearly different (Rebuttal Fig.

2h-n). Please note that the staining patterns of secreted Orion and neuronal GFP are also clearly different as originally described in Fig. 3a and b.

Rebuttal Fig. 2. Orion is secreted by MB γ axons and, once secreted, does not colocalize with the cell surface adhesion protein Fas2. a-g, 6 h APF γ axons are visualized by the expression of *201Y-GAL4*-driven *UAS-mCD8-GFP* (green) and Fas2 (red). a is a confocal Z-projection and b-g are unique confocal planes corresponding to higher magnification images of the vertical (b, d and f) and medial (c, e and g) γ lobes (rectangles in a). Note that Fas2 is

not secreted from the axons (see the high magnification image of colocalization of GFP and Fas2 in **b**, **d** and **f**) nor is observed in the “hole-like” structures (**c**, **e** and **g**). Note that the “hole-like” structures do not stain for Fas2 (arrowheads in **c**, **e** and **g**). **h-n**, 6 h APF γ axons expressing *201Y-GAL4*-driven *UAS-orion-B-myc* (red) and Fas2 (green). **h** is a confocal Z-projection and **i-n** are unique confocal planes corresponding to higher magnification images of the vertical (**i**, **k** and **m**) and medial (**j**, **l** and **n**) γ lobes (rectangles in **h**). Note that Orion, but not Fas2, is observed outside of the axons, at the tip of the vertical lobe (arrows in **i**, **k** and **m**) and in the “hole-like” structures (arrowheads in **j**, **l** and **n**). Note that, Orion is not co-localized with Fas2 in the “hole-like” structures, Scale bars represent 30 μ m. Genotypes are: $y w^{67c23} / Y$ or $y w^{67c23} / y w^*$; *UAS-mCD8GFP 201Y-GAL4* / + in **a** (n = 10) and $y w^{67c23} / Y$ or $y w^{67c23} / y w^*$; *201Y-GAL4* / +; *UAS-orion-B-myc* / + in **h** (n = 10).

If the Orion is actually secreted MB lobe, overexpression of the Myc-tagged Orion would be observed outside of MB lobe before engulfment of astrocytes. At 0h APF, when original orion is not expressed in MB neuron, these exogenous expressed Myc-Orion might be found outside of MB lobe if this molecule is actually secreted from MB lobe. Also, it is feasible to do other experiment to observe secretion of Myc-Orion; Because expression of EcR-DN in MB neurons suppresses axon pruning and successive phagocytosis by astrocytes, it is possible to distinguish the secreted Myc signals from phagocytosed Myc signals, if EcR-DN and Myc-Orion are expressed together in MB neurons.

We performed two additional experiments to address this reviewer’s comment (Rebuttal Fig.3 and 4). Indeed, a low level of Orion secretion is observable in both situations (0 h APF in wild-type and 6 h APF in *Hr39^{C13}* MBs) but not as much as in wild-type MBs at 6 h APF (Fig. 3a). This possibly indicates that the cellular machinery necessary for full Orion secretion is under the control of EcR-B1. Presumably, insufficient EcR-B1 signaling, to facilitate Orion secretion, occurs at 0 h APF in wild type animals and in EcR-B1 defective animals at 6 h APF.

Rebuttal Fig 3. Little Orion secretion is observed at 0 h APF in wild type. a-c, Confocal plane showing expression of *UAS-mCD8-GFP* (green) and *UAS-orion-B-myc* (red) under the control of *201Y-GAL4* in the vertical γ axon bundle at 0 h APF (n = 6). Note a low presence

of Myc-labelled Orion-B outside of the γ axon bundle (arrows) and inside of the vertical axon bundle (arrowheads). Scale bar represent 30 μm . Full genotype: $y w^{67c23} / Y$ or $y w^{67c23} / y w^*$; $UAS-mCD8GFP 201Y-GAL4 / +$; $UAS-orion-B-myc / +$.

Rebuttal Fig 4. Little Orion secretion is observed at 6 h APF when EcR-B1 expression is blocked in γ axons. a-c, Confocal plane showing expression of $UAS-mCD8-GFP$ (green) and $UAS-orion-B-myc$ (red) under the control of $201Y-GAL4$ in $Hr39^{C13}$ at 0 h APF (n = 10). Note a low presence of Myc-labelled Orion-B outside of the vertical γ axon bundle (arrows). Scale bar represent 30 μm . Full genotype: $y w^{67c23} / Y$ or $y w^{67c23} / y w^{67c23}$; $Hr39^{C13}$, $UAS-mCD8GFP 201Y-GAL4 / +$.

3. Using the clonal analysis of orion mutant MB neurons with MARCM, the authors showed that loss of orion in MB neurons are normally cleared by phagocyte. Their interpretation on this result is that the orion $-/+$ (heterozygote mutant) MB axons secret Orion normally and recruit phagocyte to the bundle of MB axons, resulting in phagocytosis of all MB axons by infiltrated phagocyte processes. This result indicates that Orion functions to recruit glial membrane rather than as "eat-me" signals for phagocyte. This is interesting and essential point of this research. To confirm the functional aspect of Orion, they need to perform other experiments. If pruning defect in orion mutant is induced by failure of recruitment of glial membrane, ectopic expression of orion in MB α'/β' neurons would rescue the pruning defect. MB α'/β' neurons extend their axons into the core of larval MB lobes and their axons are localized close to axons of MB gamma neurons.

This would, indeed, be very good experiment to perform. $c739-GAL4$ line is expressed strongly in the $\alpha'\beta'$ larval axons but also weakly in the γ larval axons (Fushima and Tsujimura, *Develop Growth Differ* (2007) **49**, 215-227). $orion^{\Delta C}$; $c739-GAL4/+$; $82G02/UAS-orion-B-myc$ MBs were strongly rescued (n = 116). In order to evaluate the Reviewer's hypothesis, we employed the adult-specific $\alpha'\beta'$ MB neurons $c305-GAL4$ driver line (Aso et al., *J Neurogenet* 2009). Unfortunately, $c305a-GAL4$ does not label adequately the $\alpha'\beta'$

neurons before 48 h APF (our work; Marquilly et al., submitted) and consequently the lack of rescue in *orion*^{ΔC}; *c305a-GAL4/+*; *82G02/ UAS-orion-B-myc* (n = 48) cannot be interpreted.

4. It is very important to know generality of function of Orion for recruitment of glial processes. In adult fly, severed axons are phagocytosed by glial cells. Marc Freeman's group established nice model system to observe the glial function for clearance of severed axons and published a lot of papers. Interestingly, Drpr is essential for clearance of both developmentally degenerating axons and severed axons. Is Orion also required for both pupal and adult glia to phagocyte axons? I think the authors should include this experiment in this report to further discuss function of Orion.

We will evaluate the possible roles of Orion in other aspects of neuronal remodeling. However, we feel that those studies to not be within the scope of this manuscript.

Minor points:

1. The authors revealed that "our genetic interaction analyses support orion being genetically downstream of EcR-B1." It is hard to understand the result of "genetic interaction analysis." They showed that the overexpression of EcR-B1 in orion homozygous mutant MB neurons did not affect the MB pruning phenotype of the orion mutant. I can't understand that why this result show that orion is downstream of EcR-B1. I can't accept this logic. To show this notion, the authors should perform other experiments. If the EcR-B1 signaling induces expression orion, it is expected that the overexpression of EcR-B1 in MB neurons induces more orion expression. In this case, effect of expression of orion RNAi in MB neurons would be suppressed in some extent by the increased expression of orion.

Again, a very good idea. *UAS-orion-RNAi* alone, *UAS-orion-RNAi + UAS-EcR-B1* and *UAS-orion-RNAi + UAS-FRT-y+-FRT*, as a control for the number of UASs, are described in Supplementary Fig. 3h. Overexpression of *EcR-B1* rescues the MB *orion*-RNAi remodeling phenotype. This is described in the text and we have also better explained why our results support orion being downstream of EcR-B1 (lines 146-151).

2. Although the authors measure the signal intensity of Orion-B-Myc along the MB vertical lobe and compared with that of Orion-B-deltaSP-Myc. This reviewer can't understand the point of this experiment. I can understand the requirement of signal peptide. The authors claimed the attraction of astrocytic process by Orion with this experiment. However, it is hard to understand the logic.

We captured images only when an astrocyte is present at the top of vertical lobe and, in both cases (Orion-B-Myc and Orion-B-ΔSP-Myc), in a wild-type brain. Wild-type Orion-Myc accumulates near the astrocyte (Fig. 4g and i). Orion-ΔSP-Myc is randomly distributed (Fig.

4h and j). We conclude from these results that secreted wild-type Orion, but not Δ SP Orion, activates the glial infiltration pathway.

Reviewers' Comments:

Reviewer #2:

Remarks to the Author:

The mushroom body gamma neurons are the best known model for looking at the mechanisms for controlled axon pruning in *Drosophila*. One key missing piece, though, has been the identification of a postulated attractive cue sent from the neurons to cause invasion by the surrounding glia. The authors provide compelling evidence that Orion is such a signal. The original submission did a good job of presenting their case and I think that they have made an even tighter paper in their response to the reviews. I am quite satisfied with how they addressed my comments.

I have one minor issue that may need some clarification. In lines 146-150: they state that the forced expression of UAS-EcR-B1 in the MBs did not rescue the Orion mutant phenotype (in Fig S6c,f) but then that this expression did rescue the unpruned axon phenotype produced by Orion RNAi (Table in Fig S3h). These are conflicting results that may have to do with the weaker effects of the Orion-RNAi versus the mutant but they need to say something about it.

Reviewer #3:

Remarks to the Author:

This reviewer finds a significant improvement in the manuscript. However, this reviewer still concerns about the interpretation of experimental results especially for the secretion of Orion and the phenotype of the Orion mutation.

(1) The authors claim that the presence of Myc-labelled Orion-B outside of the MB gamma axon bundle (arrows in Fig3d and 3g) is the evidence showing the secretion of Orion from MB gamma neurons. However, most of these signals were colocalized with weak GFP (GFP signals become visible when the brightness is increased). In the previous study, it has been shown that synaptic boutons degenerate and these degenerating neural debris are engulfed by astrocyte-like glia at 6h APF (Awasaki and Ito, 2004). Hakim et al (2012) showed that FasII or DsRed positive neuronal debris were located outside of MB gamma bundle. I think it is difficult to prove the secretion of Orion from MB neurons with this experiment.

(2) The authors also claim that the presence of Myc-labelled Orion-B inside of the hole-like structures present in the gamma axon bundle (arrowheads in Fig 3e and 3f) is the evidence showing the secretion of Orion from MB gamma neurons. However, it has been proposed that the hole-like structures are caused by the engulfment of synaptic boutons by a glial membrane and the hole-like structures are filled with the glial membrane (Hakim et al., 2012; Awasaki et al 2006). Therefore, it is natural to think that the Myc-Orion-B signals observed in the hole-like structures are the engulfed synaptic boutons rather than the secreted Myc-Orion-B molecules. In addition, Myc-labelled Orion-B inside of the hole-like structures was also colocalized with weak GFP.

(3) The authors also claim that the presence of Myc-Orion-B not associated with GFP-labeled axon membranes. Although this Myc-Orion-B was not associated with GFP, these signals were likely located inside the glial membrane as shown by Fig 4f. If this is the case, it is very difficult to distinguish whether Myc-Orion-B was derived from engulfed synaptic boutons or secreted from synapse then engulfed.

(4) This reviewer is confused with the description of "secretion". At first, the authors described that "Orion-B was seen along the MB lobe and at short distances away from the axons as visualized by anti-Myc staining (Fig3) (line 156-158). I think that "at short distances away from the axons" should mean Myc-Orion signals were not colocalized with GFP. However, as I show above, there is no clear evidence to show secreted Orion signal apart from GFP. Later, the authors

stated the possibility that "secreted Orion stay close to axon membrane" and showed colocalized Orion with membrane tagged GFP (Supplementary Fig 9, line 196-197).

I think "Orion is associated with membranes" is reasonable rather than "Orion is secreted by MB gamma axons." Because the chemokine is the secreted molecule, the authors would like to emphasize the "secretion" of Orion with experimental evidence. However, it is not necessary to emphasize the secretion, because the authors clearly show the non-cell-autonomous function of Orion for glial engulfment. I recommend toning down "the secretion of Orion" to avoid confusion of readers.

(5) In the revised manuscript, the authors emphasized that the larval astrocytes involved in the two separable processes on the axon pruning, axon fragmentation and clearance of axonal debris. Previously, it has been shown that inhibition of EcR in the larval astrocytes suppresses both axon fragmentation and debris clearance, whereas the *drpr* mutation suppresses only debris clearance. Based on the similarity of phenotypes, the authors claim that the loss of orion phenotype is similar to the inhibition of EcR in larval astrocytes. In such a situation, this reviewer cannot understand why the authors claim that Orion is essential for "the astrocyte infiltration." As shown by Tasdemir-Yilmaz and Freeman (2012), activation of ecdysone signaling through EcR in larval astrocytes is required for the transformation of larval astrocytes into phagocytes, which enable to find and engulf degenerated larval axons. Although the phenotype of loss of *drpr* is milder than that of orion, *drpr* mutation also suppressed infiltration of glial processes in MB axon bundles (Awasaki et al, 2006). Therefore, it would be reasonable to explain that Orion elicits not only astrocyte infiltration and but also engulfment.

(6) As pointed out by other reviewers, the function of orion in axon severing should be mentioned. This reviewer agrees that glial cells have just a minor function on axon severing and majority of axons are severed by the intrinsic signal of neurons. However, the authors emphasize that the unpruned axons are a characteristic feature of the orion mutant in the adult brain. Also, they showed that the expression of orion-RNAi in MB gamma neurons results the unpruned axon phenotype. In addition, Hakim et al also showed that suppression of EcR in astrocytes also causes unpruned axons in the adult brain. These results suggest that larval astrocytes promote axon fragmentation through extrinsic orion signals from gamma neurons.

This reviewer very confused by the ambiguous conclusion about the interaction between Orion and Drpr. The authors at first emphasize the difference of phenotype of the orion and *drpr* mutants. However, in the new manuscript, the authors claim that the unpruned phenotype is masked in the *drpr* mutant presumably by the difference of Fas2 expression (localization) on the unpruned axons. If so, the difference between the orion and *drpr* mutant is the localization of Fas2 on unpruned axons. In such a situation, it is difficult to accept the authors' assumption that "This suggests that Drpr is not an, or at least not the soe, Orion receptor." Although the authors claim that Orion does not induce the Drpr signaling pathway (Supplementary Fig. 10), it is also hard to understand the logic of why the results of Sup. Fig. 10 support their claim. It has been shown that the expression of Drpr in astrocyte is induced by EcR in astrocyte and independent of axonal degeneration of MB neurons in developmental MB pruning (Awasaki and Ito, 2004). As far as I know, it has not been shown that Drpr expression of larval astrocyte is induced by Drpr ligand. The sentence about the relationship between Drpr and Orion should be revised with a more clear and reasonable explanation.

(7) It has been shown that axons of larval olfactory projection neurons that extend their larval axons into MB calyx and lateral hone (LH) are also pruned. In addition, as shown in Sup. Fig. 1, dendrites of MB gamma neurons, which form MB calyx are also pruned. However, larval calyx and LH are filled with processes of larval astrocytes at the larval stage and astrocytes do not need to infiltrate to phagocytose degenerated debris. In general, exclude MB lobe, most of the larval synaptic regions (larval neuropil) are infiltrated with larval astrocytes (Tasdemir-Yilmaz and Freeman 2012; Omoto et al 2015). In such a situation, if the authors emphasize the function of Orion is to elicit astrocyte infiltration, it seems that Orion is specifically required for axonal pruning for MB gamma neurons in the larval MB lobe. However, as shown above, the authors also find that

dendrite pruning of MB calyx is also suppressed by loss of orion. Because MB calyx is already infiltrated by processes of larval astrocyte, it should cause confusion for readers if the authors claim that this defect is also caused by suppression of infiltration. If the authors have data showing that infiltration of astrocytes is also suppressed in MB calyx in the orion mutant, such evidence should be shown in this manuscript with images. If this is the case, Orion is functional not only for developmental pruning but also the development of larval astrocytes architecture. Please clarify this issue.

REVIEWER COMMENTS

Reviewer #2 (Remarks to the Author):

The mushroom body gamma neurons are the best known model for looking at the mechanisms for controlled axon pruning in *Drosophila*. One key missing piece, though, has been the identification of a postulated attractive cue sent from the neurons to cause invasion by the surrounding glia. The authors provide compelling evidence that *orion* is such a signal. The original submission did a good job of presenting their case and I think that they have made an even tighter paper in their response to the reviews. I am quite satisfied with how they addressed my comments.

I have one minor issue that may need some clarification. In lines 146-150: they state that the forced expression of *UAS-EcR-B1* in the MBs did not rescue the *orion* mutant phenotype (in Fig S6c,f) but then that this expression did rescue the unpruned axon phenotype produced by *orion* RNAi (Table in Fig S3h). These are conflicting results that may have to do with the weaker effects of the *orion*-RNAi versus the mutant but they need to say something about it.

These are not conflicting results. The *orion*¹ mutant allele produces a mutant protein without wild-type function. The result of forced expression of *UAS-EcR-B1* in the *orion*¹ background is an increase of the endogenous *orion*¹ transcript which will give rise to more mutant protein but without wild-type function. The expression of *orion* RNAi in the wild type background reduces the level of wild-type *orion* mRNA which in turn results in less than normal level of wild-type Orion protein with a clear phenotypic consequence. The result of forced expression of *UAS-EcR-B1* in the *orion*-targeting RNA interference background is an increase in the levels of endogenous wild-type *orion* transcript which titrates out RNA interference template and machinery resulting in sufficient levels of wild-type mRNA and protein to fully reverse the phenotypic effects of the RNAi. These results support our hypothesis that *orion* is a transcriptional target of EcR-B1.

Reviewer #3 (Remarks to the Author):

This reviewer finds a significant improvement in the manuscript. However, this reviewer still concerns about the interpretation of experimental results especially for the secretion of Orion and the phenotype of the *orion* mutation.

From her/his comments, the Reviewer #3 is apparently convinced that Orion proteins are produced by the neurons and act as a signal for “activating” the glial cells. Please note that here, activation is a general term which encompasses both infiltration and engulfment (please, see discussion below of comment #5). She/he agrees that Orion is associated with axon membranes but disagrees on the term “secretion” because, from previous studies, Fas2 which is a membrane bound receptor, and therefore not secreted, can also be located outside of the γ bundle as neuronal debris during the pruning process. We showed that Orion is produced by the γ neurons, concentrates in axon regions nearby the astrocytes and once secreted remains

close to axon membranes thanks to its link with the GAGs and “activates” the glial cells. We have used the term “secretion” because we observed a clear difference between wild-type Orion-Myc (Fig. 3a and b) and not secreted Δ SP Orion-Myc expression (Fig. 3c) and between the membrane-spanning Drl-Myc receptor and Orion-Myc (Sup Fig. 6g-i versus Sup Fig. 6j-l) and finally, because we observe that the expression pattern of “secreted” Orion and membrane-bound Fas2 are clearly different (Rebuttal Fig. 2). Nevertheless, we agree with this reviewer that, in the context of the MB remodeling, it can be very difficult and may be even impossible to show some freely diffusing Orion protein not associated with axon or glial membranes since all the axon debris will eventually end up being engulfed and phagocytosed by glial cells. If “secretion” has to be associated with freely diffusing protein, then we have, as recommended by the Reviewer #3, “toned down the secretion of Orion to avoid confusion of readers”. We therefore use the term “extracellularly present” or “extracellular” Orion instead of “secreted” Orion when appropriate.

(1) The authors claim that the presence of Myc-labelled Orion-B outside of the MB gamma axon bundle (arrows in Fig3d and 3g) is the evidence showing the secretion of Orion from MB gamma neurons. However, most of these signals were colocalized with weak GFP (GFP signals become visible when the brightness is increased). In the previous study, it has been shown that synaptic boutons degenerate and these degenerating neural debris are engulfed by astrocyte-like glia at 6h APF (Awasaki and Ito, 2004). Hakim et al (2012) showed that FasII or DsRed positive neuronal debris were located outside of MB gamma bundle. I think it is difficult to prove the secretion of Orion from MB neurons with this experiment.

We agree that it is difficult to prove secretion of Orion in this particular context. Nevertheless, extracellularly present Orion and Fas2 display clearly different expression pattern at the tip of the dorsal γ axon bundle when looked at the same developmental time points (Rebuttal Fig. 2h, i, k, m versus 2a, b, d, f). We have now corrected this point in the manuscript (lines: 156-159).

(2) The authors also claim that the presence of Myc-labelled Orion-B inside of the hole-like structures present in the gamma axon bundle (arrowheads in Fig 3e and 3f) is the evidence showing the secretion of Orion from MB gamma neurons. However, it has been proposed that the hole-like structures are caused by the engulfment of synaptic boutons by a glial membrane and the hole-like structures are filled with the glial membrane (Hakim et al., 2012; Awasaki et al 2006). Therefore, it is natural to think that the Myc-Orion-B signals observed in the hole-like structures are the engulfed synaptic boutons rather than the secreted Myc-Orion-B molecules. In addition, Myc-labelled Orion-B inside of the hole-like structures was also colocalized with weak GFP.

We agree with this concern but again, extracellularly present Orion and membrane-bound Fas2 display clearly different expression patterns in hole-like structures when looked at the same developmental time points (Rebuttal Fig. 2h, j, l, n versus 2a, c, e, g). We have now corrected this point in the manuscript (lines: 164 and 167).

(3) The authors also claim that the presence of Myc-Orion-B not associated with GFP-labeled axon membranes. Although this Myc-Orion-B was not associated with GFP, these signals were likely located inside the glial membrane as shown by Fig 4f. If this is the case, it is very difficult to distinguish whether Myc-Orion-B was derived from engulfed synaptic boutons or secreted from synapse then engulfed.

We agree that we cannot rule out this interpretation. We have now corrected this point in the manuscript (lines: 172-173).

(4) This reviewer is confused with the description of “secretion”. At first, the authors described that “Orion-B was seen along the MB lobe and at short distances away from the axons as visualized by anti-Myc staining (Fig3) (line 156-158). I think that “at short distances away from the axons” should mean Myc-Orion signals were not colocalized with GFP. However, as I show above, there is no clear evidence to show secreted Orion signal apart from GFP. Later, the authors stated the possibility that “secreted Orion stay close to axon membrane” and showed colocalized Orion with membrane tagged GFP (Supplementary Fig 9, line 196-197).

I think “Orion is associated with membranes” is reasonable rather than “Orion is secreted by MB gamma axons.” Because the chemokine is the secreted molecule, the authors would like to emphasize the “secretion” of Orion with experimental evidence. However, it is not necessary to emphasize the secretion, because the authors clearly show the non-cell-autonomous function of Orion for glial engulfment. I recommend toning down “the secretion of Orion” to avoid confusion of readers.

We agree that we cannot say that Orion is seen at short distances away from the axons and that is better to say that Orion is associated with the membranes. Nevertheless, Orion has been detected in extracellular vesicles, being present in biochemically-purified exosomes, from *Drosophila* cultured cells (now ref 27 in the manuscript: Thomas et al., PLoS Genetics 2018 14(9) : e1007694). This supports the hypothesis that Orion actually acts as a secreted protein (lines: 173-176). As was clarified in the previous rebuttal letter, fractalkine is a chemokine and the only member of the CX₃C mammalian family and it occurs in either a soluble or a membrane associated form. Therefore, *sensu stricto*, secretion is not necessary to define a chemokine. Nevertheless, we agree that it is not necessary to emphasize the secretion at the cell biological level since we have clearly showed the functional secretion of Orion by defining the non-cell-autonomous function of Orion for glial cell engulfment. We have corrected this point in the manuscript (lines: 156-159).

(5) In the revised manuscript, the authors emphasized that the larval astrocytes involved in the two separable processes on the axon pruning, axon fragmentation and clearance of axonal debris. Previously, it has been shown that inhibition of EcR in the larval astrocytes suppresses both axon fragmentation and debris clearance, whereas the drpr mutation suppresses

only debris clearance. Based on the similarity of phenotypes, the authors claim that the loss of orion phenotype is similar to the inhibition of EcR in larval astrocytes. In such a situation, this reviewer cannot understand why the authors claim that Orion is essential for “the astrocyte infiltration.” As shown by Tasdemir-Yilmaz and Freeman (2012), activation of ecdysone signaling through EcR in larval astrocytes is required for the transformation of larval astrocytes into phagocytes, which enable to find and engulf degenerated larval axons. Although the phenotype of loss of drpr is milder than that of orion, drpr mutation also suppressed infiltration of glial processes in MB axon bundles (Awasaki et al, 2006). Therefore, it would be reasonable to explain that Orion elicits not only astrocyte infiltration and but also engulfment.

We agree that Orion elicits both astrocyte infiltration and also engulfment of degenerated larval axons. We proposed the more general term astrocyte “activation” (see above) which encompasses both functions. We have corrected this point in the manuscript (lines: 59-60; 72-73; 186-187; 704-705; 874; 883-884).

(6) As pointed out by other reviewers, the function of orion in axon severing should be mentioned. This reviewer agrees that glial cells have just a minor function on axon severing and majority of axons are severed by the intrinsic signal of neurons. However, the authors emphasize that the unpruned axons are a characteristic feature of the orion mutant in the adult brain. Also, they showed that the expression of orion-RNAi in MB gamma neurons results the unpruned axon phenotype. In addition, Hakim et al also showed that suppression of EcR in astrocytes also causes unpruned axons in the adult brain. These results suggest that larval astrocytes promote axon fragmentation through extrinsic orion signals from gamma neurons. This reviewer very confused by the ambiguous conclusion about the interaction between Orion and Drpr. The authors at first emphasize the difference of phenotype of the orion and drpr mutants. However, in the new manuscript, the authors claim that the unpruned phenotype is masked in the drpr mutant presumably by the difference of Fas2 expression (localization) on the unpruned axons. If so, the difference between the orion and drpr mutant is the localization of Fas2 on unpruned axons. In such a situation, it is difficult to accept the authors' assumption that "This suggests that Drpr is not an, or at least not the soe, Orion receptor." Although the authors claim that Orion does not induce the Drpr signaling pathway (Supplementary Fig. 10), it is also hard to understand the logic of why the results of Sup. Fig. 10 support their claim. It has been shown that the expression of Drpr in astrocyte is induced by EcR in astrocyte and independent of axonal degeneration of MB neurons in developmental MB pruning (Awasaki and Ito, 2004). As far as I know, it has not been shown that Drpr expression of larval astrocyte is induced by Drpr ligand. The sentence about the relationship between Drpr and Orion should be revised with a more clear and reasonable explanation.

We realized that this part of the manuscript was somewhat unclear and have now rewritten all the corresponding paragraph in the manuscript (lines: 212-231). In particular, by using a different marker, mGFP instead of anti-Fas2, we were able to occasionally observe unpruned axons in *drpr^{Δ5}*. This is because the anti-Fas2 marker also labels the $\alpha\beta$ axons that masks individual unpruned γ axons.

(7) It has been shown that axons of larval olfactory projection neurons that extend their larval axons into MB calyx and lateral hone (LH) are also pruned. In addition, as shown in Sup. Fig. 1, dendrites of MB gamma neurons, which form MB calyx are also pruned. However, larval calyx and LH are filled with processes of larval astrocytes at the larval stage and astrocytes do not need to infiltrate to phagocytose degenerated debris. In general, exclude MB lobe, most of the larval synaptic regions (larval neuropil) are infiltrated with larval astrocytes (Tasdemir-Yilmaz and Freeman 2012; Omoto et al 2015). In such a situation, if the authors emphasize the function of Orion is to elicit astrocyte infiltration, it seems that Orion is specifically required for axonal pruning for MB gamma neurons in the larval MB lobe. However, as shown above, the authors also find that dendrite pruning of MB calyx is also suppressed by loss of orion. Because MB calyx is already infiltrated by processes of larval astrocyte, it should cause confusion for readers if the authors claim that this defect is also caused by suppression of infiltration. If the authors have data showing that infiltration of astrocytes is also suppressed in MB calyx in the orion mutant, such evidence should be shown in this manuscript with images. If this is the case, Orion is functional not only for developmental pruning but also the development of larval astrocytes architecture. Please clarify this issue.

We do not have data showing that infiltration of astrocytes is also suppressed in MB calyx in *orion* mutant. It is therefore possible that for that precise case, Orion is only required for glial engulfment (a function directly linked to transformation of astrocyte into phagocyte) and not in glial infiltration because the glial are already present in the calyx. We have added this possibility in the manuscript (lines: 781-783).

Reviewers' Comments:

Reviewer #3:

Remarks to the Author:

The authors have addressed all my concerns and much improved the paper. This is a very interesting paper that provides essential knowledge for anyone working with neuronal remodeling and neuron-glia interaction.